# GENESIS-V2: Inferring Unordered Object Representations without Iterative Refinement

**Martin Engelcke,** **Oiwi Parker Jones, and Ingmar Posner**
Applied AI Lab, University of Oxford, UK
{martin, oiwi, ingmar}@robots.ox.ac.uk

## Abstract

Advances in unsupervised learning of object-representations have culminated in the development of a broad range of methods for unsupervised object segmentation and interpretable object-centric scene generation. These methods, however, are limited to simulated and real-world datasets with limited visual complexity. Moreover, object representations are often inferred using RNNs which do not scale well to large images or iterative refinement which avoids imposing an unnatural ordering on objects in an image but requires the *a priori* initialisation of a fixed number of object representations. In contrast to established paradigms, this work proposes an embedding-based approach in which embeddings of pixels are clustered in a differentiable fashion using a stochastic stick-breaking process. Similar to iterative refinement, this clustering procedure also leads to randomly ordered object representations, but without the need of initialising a fixed number of clusters *a priori*. This is used to develop a new model, GENESIS-V2, which can infer a variable number of object representations without using RNNs or iterative refinement. We show that GENESIS-V2 performs strongly in comparison to recent baselines in terms of unsupervised image segmentation and object-centric scene generation on established synthetic datasets as well as more complex real-world datasets.

## 1 Introduction

Reasoning about discrete objects in an environment is foundational to how agents perceive their surroundings and act in it. For example, autonomous vehicles need to identify and respond to other road users (e.g. [1, 2]) and robotic manipulation tasks involve grasping and pushing individual objects (e.g. [3]). While supervised methods can identify selected objects (e.g. [4, 5]), it is intractable to manually collect labels for every possible object category. Furthermore, we often desire the ability to predict, or *imagine*, how a collection of objects might behave (e.g. [6]). A range of works have thus explored unsupervised segmentation and object-centric generation in recent years (e.g. [7–36]). These models are often formulated as variational autoencoders (VAEs) [37, 38] which allow the joint learning of inference and generation networks to identify objects in images and to generate scenes in an object-centric fashion (e.g. [15, 17, 28]).

Moreover, such models require a differentiable mechanism for separating objects in an image. While some works use spatial transformer networks (STNs) [39] to process crops that contain objects (e.g. [7–15]), others directly predict pixel-wise instance segmentation masks (e.g. [16–27]). The latter avoids the use of fixed-size sampling grids which are ill-suited for objects of varying size. Instead, object representations are inferred either by iteratively refining a set of randomly initialised representations (e.g. [19–24]) or by using a recurrent neural networks (RNN) (e.g. [16–18]). One particularly interesting model of the latter category is GENESIS [17] can perform both scene segmentation and generation by capturing relationships between objects with an autoregressive prior.

---

*Now affiliated with Google DeepMind.

35th Conference on Neural Information Processing Systems (NeurIPS 2021).

As noted in Novotny et al. [40], however, using RNNs for instance segmentation requires processing high-dimensional inputs in a sequential fashion which is computationally expensive and does not scale well to large images with potentially many objects. We also posit that recurrent inference is not only problematic from a computational point of view, but that it can also inhibit the learning of object representations by imposing an unnatural *ordering* on objects. In particular, we argue that this leads to different *object slots* receiving gradients of varying magnitude which provides a possible explanation for models collapsing to a single object slot during training, unless the flexibility of the model is restricted (see [18]). While iterative refinement instead infers unordered object representations, it requires the *a priori* initialisation of a fixed number of object slots even though the number of objects in an image is unknown.

In contrast, our work takes inspiration from the literature on supervised instance segmentation and adopts an *instance colouring* approach (e.g. [40–43]) in which pixel-wise embeddings—or colours—are clustered into attention masks. Typically, either a supervised learning signal is used to obtain cluster seeds (e.g. [40, 41]) or clustering is performed as a non-differentiable post-processing operation (e.g. [42, 43]). Neither of these approaches is suitable for unsupervised, end-to-end learning of segmentation masks. We hence develop an *instance colouring stick-breaking process* (IC-SBP) to cluster embeddings in a differentiable fashion. This is achieved by stochastically sampling cluster seeds from the pixel embeddings to perform a soft grouping of the embeddings into a set of randomly ordered attention masks. It is therefore possible to infer object representations both without imposing a fixed ordering or performing iterative refinement.

Inspired by GENESIS [17], we leverage the IC-SBP to develop GENESIS-V2, a novel model that learns to segment objects in images without supervision and that uses an autoregressive prior to generate scenes in an interpretable, object-centric fashion. GENESIS-V2 is comprehensively benchmarked against recent prior art [16, 17, 24] on established synthetic datasets—ObjectsRoom [44] and ShapeStacks [45]—where it performs strongly in comparison to several recent baselines. We also evaluate GENESIS-V2 on more challenging real-world images from the Sketchy [46] and the MIT-Princeton Amazon Picking Challenge (APC) 2016 Object Segmentation datasets [47], where it also achieves promising results. Code and pre-trained models are available at https://github.com/applied-ai-lab/genesis.

## 2 Related Work

Unsupervised models for learning object representations are typically formulated either as autoencoders (e.g. [7–10, 12–28]) or generative adversarial networks (GANs) (e.g. [29–36]). Typical GANs are able to generate images, but lack an associated inference mechanism and often suffer from training instabilities (see e.g. [48, 49]). A comprehensive review and discussion of the subject is provided in Greff et al. [50].

In order to infer object representations, STNs [39] can explicitly disentangle object location by cropping out a rectangular region from an input, allowing object appearance to be modelled in a canonical pose (e.g. [7–10, 12–15]). This operation, however, relies on a fixed-size sampling grid which is not well-suited if objects vary broadly in terms of scale. In addition, gradients are usually obtained via bi-linear interpolation and are therefore limited to the extent of the sampling grid which can impede training: for example, if the sampling grid does not overlap with any object, then its location cannot be updated in a meaningful way. In contrast, purely segmentation based approaches [16–27]) often use RNNs (e.g. [16–18]) or iterative refinement (e.g. [19–24]) to infer object representations from an image. Other works either use a fixed number of slots [25, 26] or group pixels in a non-differentiable fashion [27]. RNN based models need to learn a fixed strategy that sequentially attends to different regions in an image, but this imposes an unnatural ordering on objects in an image. Avoiding such a fixed ordering leads to a *routing problem*. One way to address this is by randomly initialising a set of object representations and iteratively refining them. More broadly, this is also related to Deep Set Prediction Networks [51], where a set is iteratively refined in a gradient-based fashion. The main disadvantage of iterative refinement is that it is necessary to initialise a fixed number of clusters *a priori*, even though ideally we would like the number of clusters to be input-dependent. This is directly facilitated by the proposed IC-SBP. The IC-SBP and GENESIS-V2 are in this respect also related to the Stick-Breaking VAE [52] which uses a stochastic number of latent variables, but does not attempt to explicitly capture the object-based structure of visual scenes.

Unlike some other works which learn unsupervised object representations from video sequences (e.g. [13, 14]), our work considers the more difficult task of learning such representations from individual images alone. GENESIS-V2 is most directly related to GENESIS [17] and SLOT-ATTENTION [24]. Like GENESIS, the model is formulated as a VAE to perform both object segmentation and object-centric scene generation, whereby the latter is facilitated by an autoregressive prior. Similar to SLOT-ATTENTION, the model uses a shared convolutional encoder to extract a feature map from which features are pooled via an attention mechanism to infer object representations with a random ordering. In contrast to SLOT-ATTENTION, however, the attention masks are obtained with a parameter-free clustering algorithm that does not require iterative refinement or a predefined number of clusters. Both GENESIS and SLOT-ATTENTION are only evaluated on synthetic datasets. In this work, we use synthetic datasets for quantitative benchmarking, but we also perform experiments on two more challenging real-world datasets.

## 3 GENESIS-V2

An image $\mathbf{x}$ of height $H$, width $W$, with $C$ channels, and pixel values in the interval $[0, 1]$ is considered to be a three-dimensional tensor $\mathbf{x} \in [0, 1]^{H \times W \times C}$. This work is only concerned with RGB images where $C = 3$, but other input modalities with a different number of channels could also be considered. Assigning individual pixels to object-like scene components can be formulated as obtaining a set of *object masks* $\pi \in [0, 1]^{H \times W \times K}$ with $\sum_k \pi_{i,j,k} = 1$ for all pixel coordinate tuples $(i, j)$ in an image, where $K$ is the number of scene components. Inspired by prior works (e.g. [16, 22]) and identical to Engelcke et al. [17], this is achieved by modelling the image likelihood $p_\theta(\mathbf{x}|\mathbf{z}_{1:K})$ as an SGMM of the form

$$\log p_\theta(\mathbf{x} \mid \mathbf{z}_{1:K}) = \sum_{i=1}^{H} \sum_{j=1}^{W} \sum_{c=1}^{C} \log \left( \sum_{k=1}^{K} \pi_{i,j,k}(\mathbf{z}_{1:K}) \, \mathcal{N}(\mu_{i,j,c}(\mathbf{z}_k), \, \sigma_x^2) \right). \tag{1}$$

The parameters $\theta$ of the model are learned $\sigma_x$ is a fixed standard deviation that is shared across object slots. The summation in Equation (1) implies that the likelihood is *permutation-invariant* to the order of the object representations $\mathbf{z}_{1:K}$ (see e.g. [53, 54]) provided that $\pi_{i,j,k}(\mathbf{z}_{1:K})$ is also permutation-invariant. This allows the generative model to accommodate for a variable number of object representations.

To segment objects in images and to generate synthetic images in an object-centric fashion requires the formulation of appropriate inference and generative models, i.e. $q_\phi(\mathbf{z}_{1:K} \mid \mathbf{x})$ and $p_\theta(\mathbf{x} \mid \mathbf{z}_{1:K}) \, p_\theta(\mathbf{z}_{1:K})$, respectively, where $\phi$ are also learnable parameters. In the generative model, it is necessary to model relationships between object representations to facilitate the generation of coherent scenes. Inspired by GENESIS [17], this is facilitated by an autoregressive prior

$$p_\theta(\mathbf{z}_{1:K}) = \prod_{k=1}^{K} p_\theta(\mathbf{z}_k \mid \mathbf{z}_{1:k-1}). \tag{2}$$

GENESIS uses two sets of latent variables to encode object masks and appearances separately. In contrast, GENESIS-V2 uses one set of latent variables $\mathbf{z}_{1:K}$ to encode both, which increases parameter sharing. The graphical model of GENESIS-V2 is shown next to related models in Appendix A.

While GENESIS relies on a recurrent mechanism in the inference model to predict segmentation masks, GENESIS-V2 instead infers latent variables without imposing a fixed ordering and assumes object latents $\mathbf{z}_{1:K}$ to be conditionally independent given an input image $\mathbf{x}$, i.e., $q_\phi(\mathbf{z}_{1:K}|\mathbf{x}) = \prod_k q_\phi(\mathbf{z}_k|\mathbf{x})$. Specifically, GENESIS-V2 first extracts an encoding with a deterministic UNet backbone. This encoding is used to predict a map of *semi-convolutional* pixel embeddings $\zeta \in \mathbb{R}^{H \times W \times D_\zeta}$ (see [40]). Semi-convolutional embeddings are introduced in Novotny et al. [40] to facilitate the prediction of unique embeddings for multiple objects of identical appearance. The embeddings are computed by performing an element-wise addition of pixel coordinates to two dimensions of the embeddings. In this work, we let the pixel coordinates be in the interval $[-1, 1]$ relative to the image centre. Subsequently, the IC-SBP converts the embeddings into a set of normalised *attention masks* $\mathbf{m} \in [0, 1]^{H \times W \times K}$ with $\sum_k m_{i,j,k} = 1$ via a distance kernel $\psi$. The spatial structure of the embeddings should induce the attention masks to be spatially localised, but this is not a hard constraint. In addition, we derive principled initialisations for the scaling-factor of different IC-SBP distance kernels $\psi$ in Section 3.2.

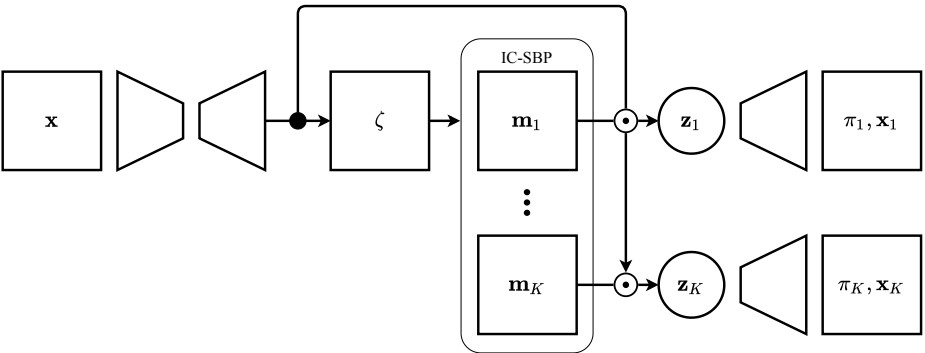

Figure 1: GENESIS-V2 overview. The image $\mathbf{x}$ is passed into a deterministic backbone. The resulting encoding is used to compute the pixel embeddings $\zeta$ which are clustered into attention masks $\mathbf{m}_{1:K}$ by the IC-SBP. Features are pooled according to these attention masks to infer the object latents $\mathbf{z}_{1:K}$. These are decoded into the object masks $\pi_{1:K}$ and reconstructed components $\mathbf{x}_{1:K}$.

Inspired by Locatello et al. [24], GENESIS-V2 uses the attention masks $\mathbf{m}_{1:K}$ to pool a feature vector for each scene component from the deterministic image encoding. A set of object latents $\mathbf{z}_{1:K}$ is then computed from these feature vectors. This set of latents is decoded in parallel to compute the statistics of the SGMM in Equation (1). The object masks $\pi$ are normalised with a softmax operator. The inference and generation models can be jointly trained as a VAE as illustrated in Figure 1. Further architecture details are described in Appendix B.

### 3.1 Instance Colouring Stick-Breaking Process

The IC-SBP is a stochastic, differentiable algorithm that clusters pixel embeddings $\zeta \in \mathbb{R}^{H \times W \times D_\zeta}$ into a variable number of soft attention masks $\mathbf{m} \in [0, 1]^{H \times W \times K}$. Intuitively, this is achieved by (1) sampling the location $(i, j)$ of a pixel that has not been assigned to a cluster yet, (2) creating a soft or hard cluster according to the distance of the embedding $\zeta_{i,j}$ at the selected pixel location to all other pixel embeddings according to a kernel $\psi$, and (3) repeating the previous two steps until all pixels are explained or some form of stopping condition is reached. Crucially, the stochastic selection of pixel embeddings as *cluster seeds* leads to a set of *randomly ordered soft clusters*. Due to its conceptual similarity, the method derives its name from more formal stick-breaking process formulations as can, e.g., be found in Yuan et al. [11] or Nalisnick and Smyth [52].

The IC-SBP is described more formally in Algorithm 1. Specifically and inspired by Burgess et al. [16], a *scope* $\mathbf{s} \in [0, 1]^{H \times W}$ is initialised to a matrix of ones $\mathbb{1}^{H \times W}$ to track the degree to which pixels have been assigned to clusters. In addition, a matrix of *seed scores* is created once by sampling from a uniform distribution $\mathbf{c} \sim U(0, 1) \in \mathbb{R}^{H \times W}$ to perform the stochastic selection of pixel embeddings. At each iteration, a single embedding vector $\zeta_{i,j}$ is selected at the spatial location $(i, j)$ which corresponds to the argmax of the element-wise multiplication of the seed scores and the current scope. This ensures that cluster seeds are sampled from pixel embeddings that have not yet been assigned to clusters. An alpha mask $\alpha_k \in [0, 1]^{H \times W}$ is computed as the distance between the cluster seed embedding $\zeta_{i,j}$ and all individual pixel embeddings according to a distance kernel $\psi$. The output of the kernel $\psi$ is one if two embeddings are identical and decreases to zero as the distance between a pair of embeddings increases. The associated attention mask $\mathbf{m}_k$ is obtained by the element-wise multiplication of the alpha masks by the current scope to ensure that the final set of attention masks is normalised. The scope is then updated by an element-wise multiplication with the complement of the alpha masks. This process is repeated until a stopping condition is satisfied, at which point the final scope is added as an additional mask to explain any remaining pixels.

In this work, we restrict ourselves to soft cluster assignment, leading to continuous attention masks with values in $\mathbf{m}_k \in [0, 1]^{H \times W}$. Unless the attention masks take binary values, several executions of the algorithm will lead to slightly different masks for individual objects. If the mask values are discrete and exactly equal to zero or one, however, then the set of cluster seeds and the set of attention masks are uniquely defined apart from their ordering. This can be inferred from the fact that if at each step of the IC-SBP produces a discrete mask, then embeddings associated with this mask cannot

---
**Algorithm 1:** Instance Colouring Stick-Breaking Process
---
**Input:** embeddings $\zeta \in \mathbb{R}^{H \times W \times D_\zeta}$
**Output:** masks $\mathbf{m}_{1:K}$ with $\mathbf{m}_k \in [0,1]^{H \times W}$
**Initialise:** masks $\mathbf{m} = \varnothing$, scope $\mathbf{s} = \mathbb{1}^{H \times W}$, seed scores $\mathbf{c} \sim U(0,1) \in \mathbb{R}^{H \times W}$

**while** *not StopCondition(*$\mathbf{m}$*)* **do**

    i, j = argmax($\mathbf{s} \odot \mathbf{c}$);
    $\alpha$ = DistanceKernel($\zeta, \zeta_{i,j}$);
    $\mathbf{m}$.append($\mathbf{s} \odot \alpha$);
    $\mathbf{s} = \mathbf{s} \odot (1 - \alpha)$;

**end**
$\mathbf{m}$.append($\mathbf{s}$)

---

be sampled as cluster seeds later on due to the masking of the seed scores by the scope. A different cluster with an associated discrete mask is therefore created at every step until all embeddings are uniquely clustered. Another interesting modification would be to use continuous masks while making the output of the IC-SBP *permutation-equivariant* with respect to the ordering of the cluster seeds. This could be achieved either by directly using the cluster seeds for downstream computations or by separating the mask normalisation from the stochastic seed selection. While the SBP formulation facilitates the selection of a diverse set of cluster seeds, the masks could be normalised separately after the cluster seeds are selected by using a softmax operation, for example. An investigation of these ideas is left for future work.

In contrast to GENESIS, the stochastic ordering of the masks implies that it is not possible for GENESIS-V2 to learn a fixed sequential decomposition strategy. While this does not strictly apply to the last mask which is set equal to the remaining scope, we find empirically that models learn a strategy where the final scope is either largely unused or where it corresponds to a generic background cluster with foreground objects remaining unordered as desired. Unlike as in iterative refinement where a fixed number of clusters needs to be initialised *a priori*, the IC-SBP can infer a variable number of object representations by using a heuristic that considers the current set of attention masks at every iteration in Algorithm 1. While we use a fixed number of $K$ masks during training for efficient parallelism on GPU accelerators, we demonstrate that a flexible number of masks can be extracted at test time with minimal impact on segmentation performance.

### 3.2 Kernel Initialisation with Semi-Convolutional Embeddings

For semi-convolutional embeddings to be similar according to a distance kernel $\psi$, the model needs to learn to compensate for the addition of the relative pixel coordinates. It can achieve this by predicting a *delta vector* for each embedding to a specific pixel location, for example to the centre of the object that the embedding belongs to. A corollary of this is that if embeddings are equal to the relative pixel coordinates with the other dimensions being zero, then clustering embeddings based on their relative distances results in "blob-like", spatially localised masks. In this work, we make use of this property to derive a meaningful initialisation for free parameters in the distance kernel $\psi$ of the IC-SBP. Established candidates for $\psi$ from the literature are the Gaussian $\psi_G$ [40], Laplacian $\psi_L$ [40], and Epanechnikov $\psi_E$ kernels [55] with

$$\psi_G = \exp\left(-\frac{||\mathbf{u} - \mathbf{v}||^2}{\sigma_G}\right), \quad \psi_L = \exp\left(-\frac{||\mathbf{u} - \mathbf{v}||}{\sigma_L}\right), \quad \psi_E = \max\left(1 - \frac{||\mathbf{u} - \mathbf{v}||^2}{\sigma_E}, 0\right), \quad (3)$$

whereby $\mathbf{u}$ and $\mathbf{v}$ are two embeddings of equal dimension. Each kernel contains a scaling factor $\sigma_{\{G,L,E\}} \in \mathbb{R}^+$. By initialising the model at the beginning of training so that the embeddings are equal to the relative pixel coordinates with the other dimensions being zero, then $\sigma_{\{G,L,E\}}$ can be initialised so that the initial attention masks are similarly-sized circular patches. In particular, we initialise these scaling factors as

$$\sigma_G^{-1} = K \ln 2, \quad \sigma_L^{-1} = \sqrt{K} \ln 2, \quad \sigma_E^{-1} = K/2, \quad (4)$$

which is derived and illustrated in Appendix C. After initialisation, $\sigma_{\{G,L,E\}}$ is jointly optimised along with the other learnable parameters of the model as in Novotny et al. [40].

## 3.3 Training

Following Engelcke et al. [17], GENESIS-V2 is trained by minimising the GECO objective [56], which can be written as a loss function of the form

$$\mathcal{L}_g = \mathbb{E}_{q_\phi(\mathbf{z}|\mathbf{x})}[-\ln p_\theta(\mathbf{x} \mid \mathbf{z})] + \beta_g \cdot \text{KL}[q_\phi(\mathbf{z} \mid \mathbf{x}) \mid\mid p_\theta(\mathbf{z})]. \qquad (5)$$

The relative weighting factor $\beta_g \in \mathbb{R}^+$ is updated at every training iteration separately from the model parameters according to

$$\beta_g = \beta_g \cdot e^{\eta(C-E)} \qquad \text{with} \qquad E = \alpha_g \cdot E + (1 - \alpha_g) \cdot \mathbb{E}_{q_\phi(\mathbf{z}|\mathbf{x})}[-\ln p_\theta(\mathbf{x} \mid \mathbf{z})]. \qquad (6)$$

$E \in \mathbb{R}$ is an exponential moving average of the negative image log-likelihood, $\alpha_g \in [0, 1]$ is a momentum factor, $\eta \in \mathbb{R}^+$ is a step size hyperparameter, and $C \in \mathbb{R}$ is a target reconstruction error. Intuitively, the optimisation decreases the weighting of the KL (Kullback-Leibler) regularisation term as long as the reconstruction error is larger than the target $C$. The weighting of the KL term is increased again once the target is satisfied.

In some applications, a practitioner might only require segmentation masks in which case, so having to reconstruct the entire input would be rather inefficient. While we observed that the attention masks $\mathbf{m}$ are correlated to the object masks $\pi$, they do not align as closely with object boundaries. We conjecture that this is a consequence of the large receptive field of the UNet backbone which spatially dilates information about objects. Consequently, we also conduct experiments with an additional auxiliary mask consistency loss that encourages attention masks $\mathbf{m}$ and object masks $\pi$ to be similar. This leads to a modified loss function of the form

$$\mathcal{L}'_g = E + \beta_g \cdot \left( \text{KL}[q_\phi(\mathbf{z} \mid \mathbf{x}) \mid\mid p_\theta(\mathbf{z})] + \text{KL}[\mathbf{m} \mid\mid \text{nograd}(\pi)] \right), \qquad (7)$$

in which $\mathbf{m}$ and $\pi$ are interpreted as pixel-wise categorical distributions. Preliminary experiments indicated that stopping the gradient propagation through the object masks $\pi$ helps to achieve segmentation quality comparable to using the original loss function in Equation (5).

## 4 Experiments

This section presents results on two simulated datasets—ObjectsRoom [44] and ShapeStacks [45]—as well as two real-world datasets—Sketchy [46] and APC [47]—which are described in Appendix D. GENESIS-V2 is compared against three recent baselines: GENESIS [17], MONET [16], and SLOT-ATTENTION [24]. Even though SLOT-ATTENTION is trained with a pure reconstruction objective and is not a generative model, it is an informative and strong baseline for unsupervised scene segmentation. The other models are trained with the GECO objective [56] following the protocol from Engelcke et al. [17] for comparability. We refer to MONET trained with GECO as MONET-G to avoid conflating the results with the original settings. Further training details are described in Appendix E.

Following prior works (e.g. [17, 18, 22, 24]), segmentation quality is quantified using the Adjusted Rand Index (ARI) [57] and the Mean Segmentation Covering (MSC). The MSC is derived from Arbelaez et al. [58] and described in detail in Engelcke et al. [17]. These are by default computed using pixels belonging to ground truth foreground objects (ARI-FG and MSC-FG). Similar to Greff et al. [22], these are averaged over 320 images from respective test sets. We also report the Evidence Lower Bound (ELBO) averaged over 320 test images as a measure how well the generative models are able to fit the data. Generation quality is measured using the Fréchet Inception Distance (FID) [59] which is computed from 10,000 samples and 10,000 test set images using the implementation from Seitzer [60]. When models are trained with multiple random seeds, we always show qualitative results for the seed that achieves the highest ARI-FG. In terms of the two real-world datasets, there are only ground truth segmentation masks available for the APC data, with the caveat that there is only a single foreground object per image. In this case, when computing the segmentation metrics from foreground pixels alone, the ARI-FG could be trivially maximised by assigning all image pixels to the same component and the MSC-FG would be equal to the largest IOU between the predicted masks and the foreground objects. When considering all pixels, the optimal solution for both metrics is to have exactly one set of pixels assigned to the foreground object and another set of pixels being assigned to the background. While acknowledging that segmenting the background as a single component is arguably not the only valid way of segmenting the background, we report the ARI and MSC using all pixels instead to develop a sense of how well foreground objects are separated from the background.

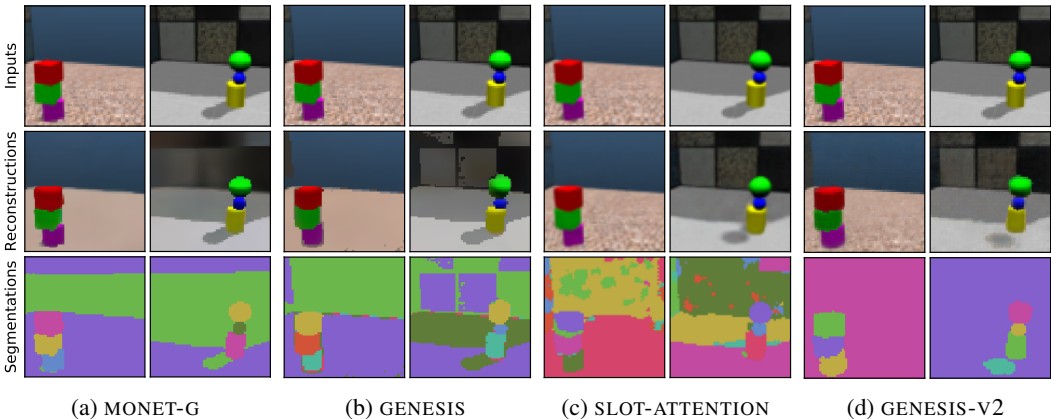

Figure 2: GENESIS-V2 learns better reconstructions and segmentations on ShapeStacks.

Table 1: Means and standard deviations of the segmentation metrics from three seeds. Bold values in the first half of the table indicate the best values for the generative models; bold values in the second half indicate any better values achieved by the additional non-generative baseline.

| Model | Generative | ObjectsRoom | | ShapeStacks | |
|---|---|---|---|---|---|
| | | ARI-FG | MSC-FG | ARI-FG | MSC-FG |
| MONET-G | Yes | 0.54±0.00 | 0.33±0.01 | 0.70±0.04 | 0.57±0.12 |
| GENESIS | Yes | 0.63±0.03 | 0.53±0.07 | 0.70±0.05 | **0.67±0.02** |
| GENESIS-V2 | Yes | **0.85±0.01** | **0.59±0.01** | **0.81±0.01** | 0.67±0.01 |
| SLOT-ATTENTION | No | 0.79±0.02 | **0.64±0.13** | 0.76±0.01 | **0.70±0.05** |

## 4.1 Benchmarking in Simulation

Figure 2 shows qualitative results on ShapeStacks and additional qualitative results are included in Appendix F. GENESIS-V2 cleanly separates the foreground objects from the background and the reconstructions. Interestingly, GENESIS-V2 is the only model that segments the entire background as a single component. We conjecture that this behaviour is a consequence of the fact that the background structures are of finite variety. As a result, even when viewing only a fraction of the background, it is possible for the model to largely predict the appearance of the rest of the background. The KL regularisation during training encourages efficient compression of the inputs, which penalises redundant information between slots and might thus explain this behaviour. SLOT-ATTENTION is trained without KL regularisation and the decoders of MONET-G as well GENESIS are possibly not flexible enough to reconstruct the entire background as a single component (see Appendix B for architecture details).

In terms of quantitative performance, Table 1 summarises the segmentation results on ObjectsRoom and ShapeStacks. GENESIS-V2 outperforms the two generative baselines GENESIS and MONET-G across datasets on all metrics, showing that the IC-SBP is indeed suitable for learning object-centric representations. GENESIS-V2 outperforms the non-generative SLOT-ATTENTION baseline in terms of the ARI-FG on both datasets. SLOT-ATTENTION manages to achieve a better mean MSC-FG. The standard deviation of the MSC-FG values is much larger, though, which indicates training is not as stable. While the ARI-FG indicates the models ability to separate objects, it does not penalise the undersegmentation of objects (see [17]). The MSC-FG, in contrast, is an IOU based metric and sensitive to the exact segmentation masks. We conjecture that SLOT-ATTENTION manages to predict slightly more accurate segmentation masks given that is trained on a pure reconstruction objective and without KL regularisation, thus leading to a slightly better mean MSC-FG. A set of ablations for GENESIS-V2 is also included in Appendix F.

Table 2: Means and standard deviations of the segmentation metrics from three seeds for GENESIS-V2 with a fixed or flexible number of object slots. Highlighting follows an analogous scheme as in Table 1.

| Dataset | Training | Slots | Avg. $K \downarrow$ | MAE $\downarrow$ | ARI-FG $\uparrow$ | MSC-FG $\uparrow$ |
|---|---|---|---|---|---|---|
| ObjectsRoom | No mask loss | Fixed | 7.0±0.0 | 3.3±0.0 | **0.85±0.01** | **0.59±0.01** |
| | | Flexible | **5.0±0.9** | **1.7±0.9** | 0.84±0.01 | 0.51±0.10 |
| ShapeStacks | No mask loss | Fixed | 9.0±0.0 | 4.4±0.0 | **0.81±0.01** | **0.67±0.01** |
| | | Flexible | **6.3±0.3** | **1.9±0.3** | 0.77±0.02 | 0.63±0.01 |
| | With mask loss | Fixed | 9.0±0.0 | 4.4±0.0 | **0.81±0.01** | **0.68±0.00** |
| | | Flexible | **5.7±0.2** | **1.1±0.02** | **0.81±0.01** | **0.68±0.01** |

Table 3: Means and standard deviations of ELBO values and FID scores from three seeds.

| Model | ObjectsRoom | | ShapeStacks | |
|---|---|---|---|---|
| | ELBO $\uparrow$ | FID $\downarrow$ | ELBO $\uparrow$ | FID $\downarrow$ |
| MONET-G | -7217±19 | 205.7±7.6 | -7268±19 | 197.8±5.2 |
| GENESIS | **-7023±2** | 62.8±2.5 | -7082±15 | 186.8±18.0 |
| SLOT-ATT. | — | — | — | — |
| GENESIS-V2 | -7040±2 | **52.6±2.7** | **-7019±2** | **112.7±3.2** |

We also examine whether the IC-SBP can indeed be used to extract a variable number of object representations. This is done by terminating the IC-SBP according to a manual heuristic and setting the final mask to the remaining scope. Specifically, we terminate the IC-SBP when the sum of the current attention mask values is smaller than 70 pixels for ObjectsRoom and 20 pixels for ShapeStacks. A larger threshold is used for the former as the attention masks tend to be more dilated (see Appendix F). The average number of used slots, the Mean Absolute Error (MAE) to the ideal number of slots, and segmentation metrics are when using a fixed number and a variable number slots after training are summarised in Table 2. On both datasets, allowing for a flexible number requires fewer steps and achieves a smaller MAE. This is incurred, though, at a drop in segmentation performance. When training GENESIS-V2 with the auxiliary mask loss as in Equation (7) on ShapeStacks, the average number of steps and the MAE further decrease at no impact on the segmentation metrics. On ObjectsRoom, the auxiliary mask loss appeared to deteriorate the learning of good object segmentations and the associated results are therefore not included.

In terms of density estimation and scene generation, Table 3 summarises the ELBO values and FID scores for GENESIS-V2 and the baselines on ObjectsRoom and ShapeStacks.[2] GENESIS achieves a slightly better ELBO than GENESIS-V2 on ObjectsRoom, but GENESIS-V2 performs significantly better on ShapeStacks. We hypothesise that GENESIS benefits here from having a more flexible, autoregressive posterior than GENESIS-V2. Regarding scene generation, GENESIS-V2 consistently performs best with a particularly significant improvement on ShapeStacks. Results on ShapeStacks, however, are not as good as for ObjectsRoom, which is likely caused by the increased visual complexity of the images in the ShapeStacks dataset. Qualitative results for scene generation are shown in Figures 3 and 4. Both GENESIS and GENESIS-V2 produce reasonable samples after training on ObjectsRoom. For ShapeStacks, samples from GENESIS contain significant distortions. In comparison, samples from GENESIS-V2 are more realistic, but also still show room for improvement.

---

[2]Note that in contrast to the MONET-G training objective, the decoded mask distribution rather than the deterministic attention masks are used to compute the reconstruction likelihood in the ELBO calculation for MONET-G. The KL divergence between the two mask distributions as used in the training objective is not part of this ELBO calculation.

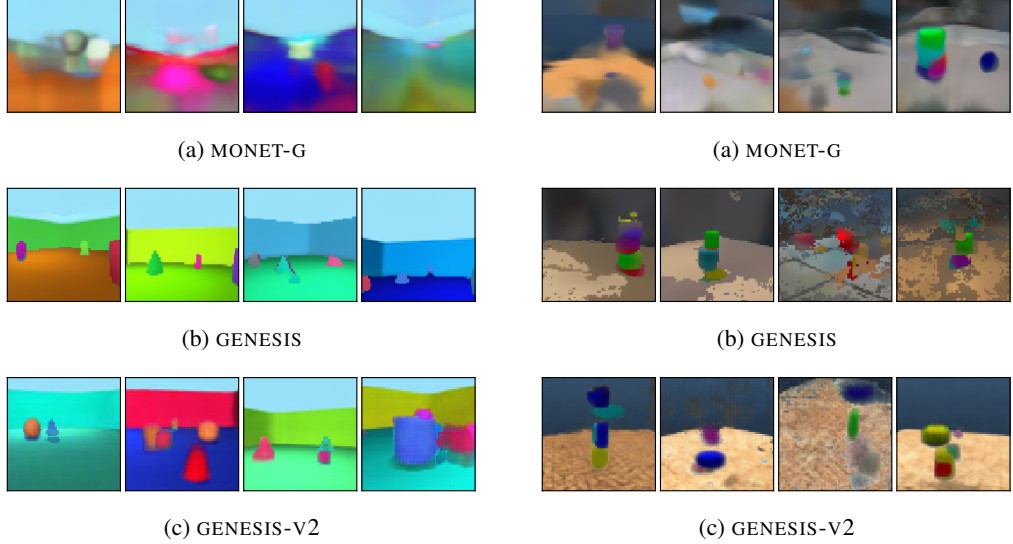

(a) MONET-G                   (a) MONET-G

(b) GENESIS                  (b) GENESIS

(c) GENESIS-V2              (c) GENESIS-V2

Figure 3: ObjectsRoom samples.        Figure 4: ShapeStacks samples.

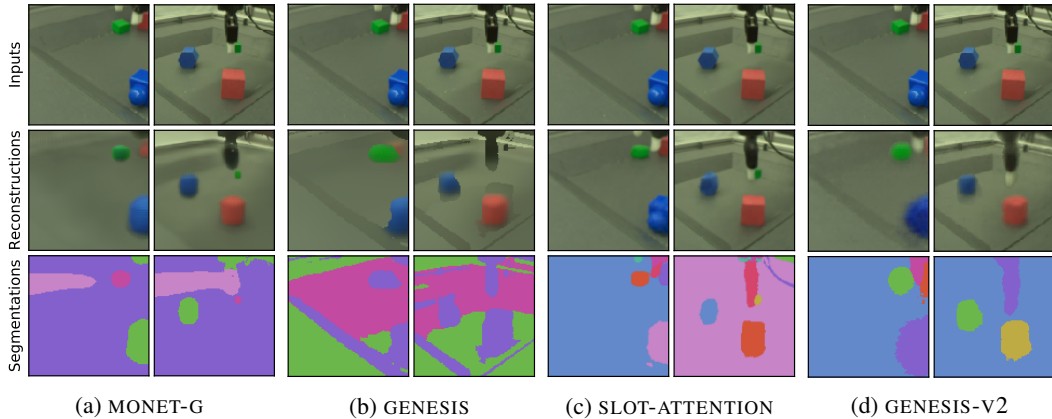

(a) MONET-G       (b) GENESIS       (c) SLOT-ATTENTION       (d) GENESIS-V2

Figure 5: In contrast to MONET-G and GENESIS, GENESIS-V2 as well as SLOT-ATTENTION are able to learn reasonable object segmentations on the more challenging Sketchy dataset.

## 4.2 Real-World Applications

After validating GENESIS-V2 on two simulated datasets, this section present experiments on the Sketchy [46] and APC datasets [47]; two significantly more challenging real-world datasets collected in the context of robot manipulation. Due to the long training time on these datasets, each model is only trained with a single random seed. While this makes it infeasible to draw statistically strong conclusions, the aim of this section is to provide an indication and early exploration of how these models fare on more complex real-world datasets. Reconstructions and segmentations after training GENESIS-V2 on Sketchy and APC are shown in Figures 5 and 6. For Sketchy, it can be seen that GENESIS-V2 and SLOT-ATTENTION disambiguate the individual foreground objects and the robot gripper fairly well. SLOT-ATTENTION produces slightly more accurate reconstructions, which is likely facilitated by the pure reconstruction objective that the model is trained with. However, GENESIS-V2 is the only model that separates foreground objects from the background in APC images. Environment conditions in Sketchy are highly controlled and SLOT-ATTENTION appears to be unable to handle the more complex conditions in APC images. Nevertheless, GENESIS-V2 also struggles to capture the fine-grained details and oversegments one of the foreground objects into several parts, leaving room for improvement in future work. Additional qualitative results are included in Appendix F.

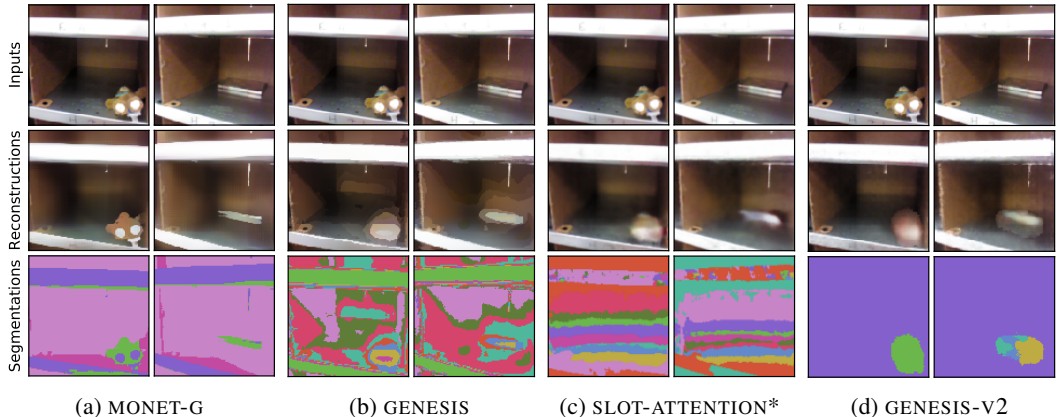

Figure 6: GENESIS-V2 is the only model that separates the foreground objects in images from APC.

Table 4: Segmentation metrics on APC.

|  | ARI | MSC |
|---|---|---|
| MONET-G | 0.11 | 0.48 |
| GENESIS | 0.04 | 0.29 |
| SLOT-ATT. | 0.03 | 0.25 |
| GENESIS-V2 | **0.55** | **0.67** |

Table 5: FID scores on Sketchy and APC.

|  | Sketchy | APC |
|---|---|---|
| MONET-G | 294.3 | 269.3 |
| GENESIS | 241.9 | **183.2** |
| SLOT-ATT. | — | — |
| GENESIS-V2 | **208.1** | 245.6 |

Table 4 reports the ARI and MSC scores computed from all pixels GENESIS-V2 and the baselines. GENESIS-V2 stands out in terms of both metrics, corroborating that GENESIS-V2 takes a valuable step towards learning unsupervised object-representations from *real-world* datasets. FID scores for generated images are summarised in Table 5 and qualitative results are included in Appendix F. GENESIS-V2 achieves the best FID on Sketchy, but it is outperformed by GENESIS on APC. Both models consistently outperform MONET-G. All of the FID scores, however, are fairly large which is not surprising given the much higher visual complexity of these images. It is therefore difficult to draw strong conclusions from these beyond a rough sense of sample quality. Further work is required to generate high-fidelity images after training on real-world datasets.

## 5 Conclusions

This work develops GENESIS-V2, a novel object-centric latent variable model of scenes which is able to both decompose visual scenes into semantically meaningful constituent parts while at the same time being able to generate coherent scenes in an object-centric fashion. GENESIS-V2 leverages a differentiable clustering algorithm for grouping pixel embeddings into a variable number of attention masks which are used to infer an unordered set of object representations. This approach is validated empirically on two established simulated datasets as well as two additional real-world datasets. The results show that GENESIS-V2 takes a step towards learning better object-centric representations without labelled supervision from real-world datasets.

In terms of future work, there is still room for improvement in terms of reconstruction, segmentation, and sample quality. It would also be interesting to investigate the ability of GENESIS-V2 and the IC-SBP to handle out-of-distribution images that contain more objects than seen during training. Moreover, several interesting variations of the IC-SBP were also already described in Section 3.1. Additional promising avenues include the extension of GENESIS-V2 to learn object representations from video (see e.g. [13, 14]) or to leverage recent advances in hierarchical latent variable models such as Nouveau VAEs (NVAEs) [61].

