## Acknowledgments and Disclosure of Funding

This research was supported by an EPSRC Programme Grant (EP/V000748/1) and a gift from Amazon Web Services (AWS). The authors would like to acknowledge the use of the University of Oxford Advanced Research Computing (ARC) facility in carrying out this work, `http://dx.doi.org/10.5281/zenodo.22558`, and the use of Hartree Centre resources. The Oxford Robotics Institute is supported by SCAN Computers in the form of hardware and services. Martin Engelcke was funded by an EPSRC DTA Studentship and a Google Studentship during his doctoral studies. Finally, the authors thank Yizhe Wu and Adam R. Kosiorek for useful comments and the NeurIPS reviewers for their time and feedback.

## References

[1] Andreas Geiger, Philip Lenz, and Raquel Urtasun. Are We Ready for Autonomous Driving? The KITTI Vision Benchmark Suite. In *IEEE/CVF Conference on Computer Vision and Pattern Recognition (CVPR)*, 2012.

[2] Marius Cordts, Mohamed Omran, Sebastian Ramos, Timo Rehfeld, Markus Enzweiler, Rodrigo Benenson, Uwe Franke, Stefan Roth, and Bernt Schiele. The Cityscapes Dataset for Semantic Urban Scene Understanding. In *IEEE/CVF Conference on Computer Vision and Pattern Recognition (CVPR)*, 2016.

[3] Coline Devin, Pieter Abbeel, Trevor Darrell, and Sergey Levine. Deep Object-Centric Representations for Generalizable Robot Learning. In *IEEE International Conference on Robotics and Automation (ICRA)*, 2018.

[4] Shaoqing Ren, Kaiming He, Ross Girshick, and Jian Sun. Faster R-CNN: Towards Real-Time Object Detection with Region Proposal Networks. In *Advances in Neural Information Processing Systems (NeurIPS)*, 2015.

[5] Kaiming He, Georgia Gkioxari, Piotr Dollár, and Ross Girshick. Mask R-CNN. In *International Conference on Computer Vision (ICCV)*, 2017.

[6] Bohan Wu, Suraj Nair, Roberto Martin-Martin, Li Fei-Fei, and Chelsea Finn. Greedy Hierarchical Variational Autoencoders for Large-Scale Video Prediction. In *IEEE/CVF Conference on Computer Vision and Pattern Recognition (CVPR)*, 2021.

[7] Jonathan Huang and Kevin Murphy. Efficient Inference in Occlusion-Aware Generative models of Images. *arXiv preprint arXiv:1511.06362*, 2015.

[8] SM Ali Eslami, Nicolas Heess, Theophane Weber, Yuval Tassa, David Szepesvari, Geoffrey E Hinton, et al. Attend, Infer, Repeat: Fast Scene Understanding with Generative Models. In *Advances in Neural Information Processing Systems (NeurIPS)*, 2016.

[9] Kun Xu, Chongxuan Li, Jun Zhu, and Bo Zhang. Multi-Objects Generation with Amortized Structural Regularization. In *Advances in Neural Information Processing Systems (NeurIPS)*, 2018.

[10] Eric Crawford and Joelle Pineau. Spatially Invariant Unsupervised Object Detection with Convolutional Neural Networks. In *AAAI Conference on Artificial Intelligence*, 2019.

[11] Jinyang Yuan, Bin Li, and Xiangyang Xue. Generative Modeling of Infinite Occluded Objects for Compositional Scene Representation. In *International Conference on Machine Learning (ICML)*, 2019.

[12] Zhixuan Lin, Yi-Fu Wu, Skand Vishwanath Peri, Weihao Sun, Gautam Singh, Fei Deng, Jindong Jiang, and Sungjin Ahn. SPACE: Unsupervised Object-Oriented Scene Representation via Spatial Attention and Decomposition. In *International Conference on Learning Representations (ICLR)*, 2020.

[13] Adam Kosiorek, Hyunjik Kim, Yee Whye Teh, and Ingmar Posner. Sequential Attend, Infer, Repeat: Generative Modelling of Moving Objects. In *Advances in Neural Information Processing Systems (NeurIPS)*, 2018.

[14] Jindong Jiang, Sepehr Janghorbani, Gerard De Melo, and Sungjin Ahn. SCALOR: Generative World Models with Scalable Object Representations. In *International Conference on Learning Representations (ICLR)*, 2020.

[15] Jindong Jiang and Sungjin Ahn. Generative Neurosymbolic Machines. In *Advances in Neural Information Processing Systems (NeurIPS)*, 2020.

[16] Christopher P Burgess, Loic Matthey, Nicholas Watters, Rishabh Kabra, Irina Higgins, Matt Botvinick, and Alexander Lerchner. MONet: Unsupervised Scene Decomposition and Representation. *arXiv preprint arXiv:1901.11390*, 2019.

[17] Martin Engelcke, Adam R Kosiorek, Oiwi Parker Jones, and Ingmar Posner. GENESIS: Generative Scene Inference and Sampling with Object-Centric Latent Representations. In *International Conference on Learning Representations (ICLR)*, 2020.

[18] Martin Engelcke, Oiwi Parker Jones, and Ingmar Posner. Reconstruction Bottlenecks in Object-Centric Generative Models. *ICML Workshop on Object-Oriented Learning*, 2020.

[19] Klaus Greff, Antti Rasmus, Mathias Berglund, Tele Hao, Harri Valpola, and Jürgen Schmidhuber. Tagger: Deep Unsupervised Perceptual Grouping. In *Advances in Neural Information Processing Systems (NeurIPS)*, 2016.

[20] Klaus Greff, Sjoerd van Steenkiste, and Jürgen Schmidhuber. Neural Expectation Maximization. In *Advances in Neural Information Processing Systems (NeurIPS)*, 2017.

[21] Sjoerd van Steenkiste, Michael Chang, Klaus Greff, and Jürgen Schmidhuber. Relational Neural Expectation Maximization: Unsupervised Discovery of Objects and their Interactions. In *International Conference on Learning Representations (ICLR)*, 2018.

[22] Klaus Greff, Raphaël Lopez Kaufmann, Rishab Kabra, Nick Watters, Chris Burgess, Daniel Zoran, Loic Matthey, Matthew Botvinick, and Alexander Lerchner. Multi-Object Representation Learning with Iterative Variational Inference. In *International Conference on Machine Learning (ICML)*, 2019.

[23] Rishi Veerapaneni, John D Co-Reyes, Michael Chang, Michael Janner, Chelsea Finn, Jiajun Wu, Joshua Tenenbaum, and Sergey Levine. Entity Abstraction in Visual Model-Based Reinforcement Learning. In *Conference on Robot Learning (CoRL)*, 2020.

[24] Francesco Locatello, Dirk Weissenborn, Thomas Unterthiner, Aravindh Mahendran, Georg Heigold, Jakob Uszkoreit, Alexey Dosovitskiy, and Thomas Kipf. Object-Centric Learning with Slot Attention. In *Advances in Neural Information Processing Systems (NeurIPS)*, 2020.

[25] Adam R Kosiorek, Sara Sabour, Yee Whye Teh, and Geoffrey E Hinton. Stacked Capsule Autoencoders. In *Advances in Neural Information Processing Systems (NeurIPS)*, 2019.

[26] Yanchao Yang, Yutong Chen, and Stefano Soatto. Learning to Manipulate Individual Objects in an Image. In *IEEE/CVF Conference on Computer Vision and Pattern Recognition (CVPR)*, 2020.

[27] Daniel M Bear, Chaofei Fan, Damian Mrowca, Yunzhu Li, Seth Alter, Aran Nayebi, Jeremy Schwartz, Li Fei-Fei, Jiajun Wu, Joshua B Tenenbaum, et al. Learning Physical Graph Representations from Visual Scenes. In *Advances in Neural Information Processing Systems (NeurIPS)*, 2020.

[28] Titas Anciukevicius, Christoph H Lampert, and Paul Henderson. Object-Centric Image Generation with Factored Depths, Locations, and Appearances. *arXiv preprint arXiv:2004.00642*, 2020.

[29] Sjoerd van Steenkiste, Karol Kurach, and Sylvain Gelly. A Case for Object Compositionality in Deep Generative Models of Images. *NeurIPS Workshop on Modeling the Physical World: Learning, Perception, and Control*, 2018.

[30] Mickaël Chen, Thierry Artières, and Ludovic Denoyer. Unsupervised Object Segmentation by Redrawing. In *Advances in Neural Information Processing Systems (NeurIPS)*, 2019.

[31] Adam Bielski and Paolo Favaro. Emergence of Object Segmentation in Perturbed Generative Models. In *Advances in Neural Information Processing Systems (NeurIPS)*, 2019.

[32] Relja Arandjelović and Andrew Zisserman. Object Discovery with a Copy-Pasting GAN. *arXiv preprint arXiv:1905.11369*, 2019.

[33] Samaneh Azadi, Deepak Pathak, Sayna Ebrahimi, and Trevor Darrell. Compositional GAN: Learning Image-Conditional Binary Composition. *arXiv preprint arXiv:1807.07560*, 2019.

[34] Thu Nguyen-Phuoc, Christian Richardt, Long Mai, Yong-Liang Yang, and Niloy Mitra. BlockGAN: Learning 3D Object-aware Scene Representations from Unlabelled Images. In *Advances in Neural Information Processing Systems (NeurIPS)*, 2020.

[35] Sebastien Ehrhardt, Oliver Groth, Aron Monszpart, Martin Engelcke, Ingmar Posner, Niloy Mitra, and Andrea Vedaldi. RELATE: Physically Plausible Multi-Object Scene Synthesis Using Structured Latent Spaces. In *Advances in Neural Information Processing Systems (NeurIPS)*, 2020.

[36] Michael Niemeyer and Andreas Geiger. GIRAFFE: Representing Scenes as Compositional Generative Neural Feature Fields. In *IEEE/CVF Conference on Computer Vision and Pattern Recognition (CVPR)*, 2020.

[37] Diederik P Kingma and Max Welling. Auto-Encoding Variational Bayes. In *International Conference on Learning Representations (ICLR)*, 2014.

[38] Danilo Jimenez Rezende, Shakir Mohamed, and Daan Wierstra. Stochastic Backpropagation and Approximate Inference in Deep Generative Models. In *International Conference on Machine Learning (ICML)*, 2014.

[39] Max Jaderberg, Karen Simonyan, Andrew Zisserman, and Koray Kavukcuoglu. Spatial Transformer Networks. In *Advances in Neural Information Processing Systems (NeurIPS)*, 2015.

[40] David Novotny, Samuel Albanie, Diane Larlus, and Andrea Vedaldi. Semi-Convolutional Operators for Instance Segmentation. In *European Conference on Computer Vision (ECCV)*, 2018.

[41] Alireza Fathi, Zbigniew Wojna, Vivek Rathod, Peng Wang, Hyun Oh Song, Sergio Guadarrama, and Kevin P Murphy. Semantic Instance Segmentation via Deep Metric Learning. *arXiv preprint arXiv:1703.10277*, 2017.

[42] Min Bai and Raquel Urtasun. Deep Watershed Transform for Instance Segmentation. In *IEEE/CVF Conference on Computer Vision and Pattern Recognition (CVPR)*, 2017.

[43] Bert De Brabandere, Davy Neven, and Luc Van Gool. Semantic Instance Segmentation with a Discriminative Loss Function. *arXiv preprint arXiv:1708.02551*, 2017.

[44] Rishabh Kabra, Chris Burgess, Loic Matthey, Raphael Lopez Kaufman, Klaus Greff, Malcolm Reynolds, and Alexander Lerchner. Multi-Object Datasets, 2019. URL https://github.com/deepmind/multi-object-datasets/.

[45] Oliver Groth, Fabian B Fuchs, Ingmar Posner, and Andrea Vedaldi. ShapeStacks: Learning Vision-Based Physical Intuition for Generalised Object Stacking. In *European Conference on Computer Vision (ECCV)*, 2018.

[46] Serkan Cabi, Sergio Gómez Colmenarejo, Alexander Novikov, Ksenia Konyushkova, Scott Reed, Rae Jeong, Konrad Zolna, Yusuf Aytar, David Budden, Mel Vecerik, et al. Scaling Data-Driven Robotics with Reward Sketching and Batch Reinforcement Learning. In *Robotics: Science and Systems (RSS)*, 2020.

[47] Andy Zeng, Kuan-Ting Yu, Shuran Song, Daniel Suo, Ed Walker Jr, Alberto Rodriguez, and Jianxiong Xiao. Multi-view Self-supervised Deep Learning for 6D Pose Estimation in the Amazon Picking Challenge. In *IEEE International Conference on Robotics and Automation (ICRA)*, 2017.

[48] Ian Goodfellow, Jean Pouget-Abadie, Mehdi Mirza, Bing Xu, David Warde-Farley, Sherjil Ozair, Aaron Courville, and Yoshua Bengio. Generative Adversarial Nets. In *Advances in Neural Information Processing Systems (NeurIPS)*, 2014.

[49] Andrew Brock, Jeff Donahue, and Karen Simonyan. Large Scale GAN Training for High Fidelity Natural Image Synthesis. In *International Conference on Learning Representations (ICLR)*, 2019.

[50] Klaus Greff, Sjoerd van Steenkiste, and Jürgen Schmidhuber. On the Binding Problem in Artificial Neural Networks. *arXiv preprint arXiv:2012.05208*, 2020.

[51] Yan Zhang, Jonathon Hare, and Adam Prügel-Bennett. Deep Set Prediction Networks. In *Advances in Neural Information Processing Systems (NeurIPS)*, 2019.

[52] Eric Nalisnick and Padhraic Smyth. Stick-Breaking Variational Autoencoders. In *International Conference on Learning Representations (ICLR)*, 2017.

[53] Manzil Zaheer, Satwik Kottur, Siamak Ravanbakhsh, Barnabas Poczos, Ruslan Salakhutdinov, and Alexander Smola. Deep Sets. In *Advances in Neural Information Processing Systems (NeurIPS)*, 2017.

[54] Edward Wagstaff, Fabian Fuchs, Martin Engelcke, Ingmar Posner, and Michael A Osborne. On the Limitations of Representing Functions on Sets. In *International Conference on Machine Learning (ICML)*, 2019.

[55] Shu Kong and Charless C Fowlkes. Recurrent Pixel Embedding for Instance Grouping. In *IEEE/CVF Conference on Computer Vision and Pattern Recognition (CVPR)*, 2018.

[56] Danilo Jimenez Rezende and Fabio Viola. Taming VAEs. *arXiv preprint arXiv:1810.00597*, 2018.

[57] Lawrence Hubert and Phipps Arabie. Comparing Partitions. *Journal of Classification*, 2(1):193–218, 1985.

[58] Pablo Arbelaez, Michael Maire, Charless Fowlkes, and Jitendra Malik. Contour Detection and Hierarchical Image Segmentation. *IEEE Transactions on Pattern Analysis and Machine Intelligence*, 33(5), 2010.

[59] Martin Heusel, Hubert Ramsauer, Thomas Unterthiner, Bernhard Nessler, and Sepp Hochreiter. GANs Trained by a Two Time-Scale Update Rule Converge to a Local Nash Equilibrium. In *Advances in Neural Information Processing Systems (NeurIPS)*, 2017.

[60] Maximilian Seitzer. pytorch-fid: FID Score for PyTorch. https://github.com/mseitzer/pytorch-fid, August 2020. Version 0.1.1.

[61] Arash Vahdat and Jan Kautz. NVAE: A Deep Hierarchical Variational Autoencoder. In *Advances in Neural Information Processing Systems (NeurIPS)*, 2020.

[62] Olaf Ronneberger, Philipp Fischer, and Thomas Brox. U-Net: Convolutional Networks for Biomedical Image Segmentation. In *International Conference on Medical Image Computing and Computer-Assisted Intervention (MICCAI)*, 2015.

[63] Dmitry Ulyanov, Andrea Vedaldi, and Victor Lempitsky. Instance Normalization: The Missing Ingredient for Fast Stylization. *arXiv preprint arXiv:1607.08022*, 2016.

[64] Yuxin Wu and Kaiming He. Group Normalization. In *European Conference on Computer Vision (ECCV)*, 2018.

[65] Jimmy Lei Ba, Jamie Ryan Kiros, and Geoffrey E Hinton. Layer Normalization. *arXiv preprint arXiv:1607.06450*, 2016.

[66] Nicholas Watters, Loic Matthey, Christopher P Burgess, and Alexander Lerchner. Spatial Broadcast Decoder: A Simple Architecture for Learning Disentangled Representations in VAEs. *ICLR Workshop on Learning from Limited Data*, 2019.

[67] Sepp Hochreiter and Jürgen Schmidhuber. Long Short-Term Memory. *Neural Computation*, 1997.

[68] Diederik P Kingma and Jimmy Ba. Adam: A Method for Stochastic Optimization. In *International Conference on Learning Representations (ICLR)*, 2015.

[69] Adam Paszke, Sam Gross, Soumith Chintala, Gregory Chanan, Edward Yang, Zachary DeVito, Zeming Lin, Alban Desmaison, Luca Antiga, and Adam Lerer. Automatic Differentiation in PyTorch. 2017.


# A  Graphical Models

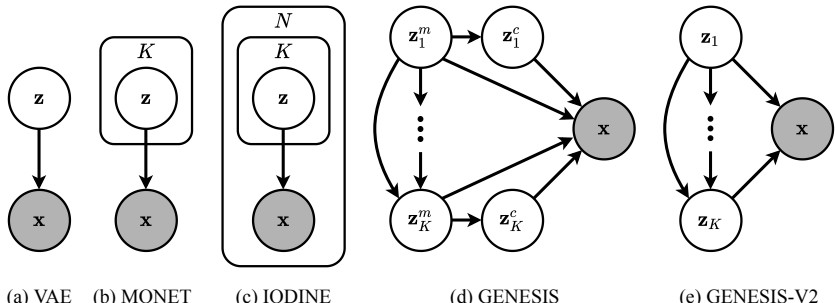

Figure 7: Graphical model of GENESIS-V2 compared to a standard VAE [37, 38], MONET [16], IODINE [22], and GENESIS [17]. $N$ denotes the number of refinement iterations in IODINE. GENESIS and GENESIS-V2 capture correlations between object slots with an autoregressive prior.

# B  GENESIS-V2 Architecture Details

The GENESIS-V2 architecture consists of four main components: a deterministic backbone, the attention and object pooling module, the component decoders, and an optional autoregressive prior which are described in detail below.

**Backbone**  GENESIS-V2 uses a UNet [62] encoder similar to the attention network in the re-implementation of MONET in Engelcke et al. [17] with $[64, 64, 128, 128, 128]$ filters in the encoder and the reverse in the decoder. Each convolutional block decreases or increases the spatial resolution by a factor of two and there are two hidden layers with 128 units each in between the encoder and the decoder. The only difference to the UNet implementation in Engelcke et al. [17] is that the instance normalisation (IN) layers [63] are replaced with group normalisation (GN) layers [64] to preserve contrast information. The number of groups is set to eight in all such layers which is also referred to as a GN8 layer. The output of this backbone encoder is a feature map $\mathbf{e} \in \mathbb{R}^{H \times W \times D_e}$ with $D_e = 64$ output channels and spatial dimensions that are equal to the height and width of the input image.

**Attention and Object Pooling**  Following feature extraction, an *attention head* computes pixel-wise semi-convolutional embeddings $\zeta$ with eight channels, i.e. $D_\zeta = 8$, as in Novotny et al. [40]. The attention head consists of a $3 \times 3$ Conv-GN8-ReLU block with 64 filters and a $1 \times 1$ semi-convolutional layer. The pixel embeddings are clustered into $K$ attention masks $\mathbf{m}_{1:K}$ using the IC-SBP. A Gaussian kernel $\psi_G$ is used unless noted otherwise. A *feature head* consisting of a $3 \times 3$ Conv-GN8-ReLU block with 64 filters and a $1 \times 1$ convolution with 128 filters refines the encoder output $\mathbf{e}$ to obtain a new feature map $\mathbf{f} \in \mathbb{R}^{H \times W \times D_f}$ with $D_f = 128$. Similar to Locatello et al. [24], the attention masks $\mathbf{m}_{1:K}$ are used to pool feature vectors from the feature map by multiplying the feature map with an individual attention mask and summing across the spatial dimensions. Each pooled feature vector is normalised by dividing by the sum of the attention mask values plus a small epsilon value to avoid numerical instabilities. Finally, a *posterior head* uses layer normalisation [65] followed by a fully-connected ReLU block with 128 units and a second fully-connected layer to compute the sufficient statistics of the individual object latents $\mathbf{z}_{1:K}$ with $\mathbf{z}_k \in \mathbb{R}^{64}$ from pooled feature vector.

**Component Decoders**  Following Greff et al. [22] and Locatello et al. [24], the object latents are decoded by separate decoders with shared weights to parameterise the sufficient statistics of the SGMM in Equation (1). Each decoded component has four channels per pixel. The first three channels contain the RGB values and the fourth channel contains the unnormalised segmentation logits which are normalised across scene components using a softmax operator. Again following Locatello et al. [24], the first layer is a spatial broadcasting module as introduced in Watters et al. [66] which is designed to facilitate the disentanglement of the independent factors of variation in a dataset. An additional advantage of spatial broadcasting is that it requires a smaller number of parameters than a fully-connected layer when upsampling a feature vector to a specific spatial resolution. The

spatial broadcasting module is followed by four $5 \times 5$, stride-2 deconvolutional GN8-ReLU layers with 64 filters to retrieve the full image resolution before a final $1 \times 1$ convolution which computes the four output channels. The use of stride-2 deconvolutional layers should make the GENESIS-V2 decoder more flexible compared to the counterparts used in MONET-G and GENESIS, which broadcast higher resolution and use stride-1 convolutions for decoding (see also [18]).

**Autoregressive Prior** Identical to GENESIS [17], the autoregressive prior for scene generation is implemented as an LSTM [67] followed by a fully-connected linear layer with 256 units to infer the sufficient statistics of the prior distribution for each component.

## C   Kernel Initialisation

Assume a maximum of $K$ scene components to be present in an image and that model is initialised so that the pixel embeddings are equal to the relative pixel coordinates with the other dimensions being zero at the beginning of training. For each initial mask to cover approximately the same area of an image, further assume that the circular isocontours of the kernels are packed into an image in a square fashion. Using linear relative pixel coordinates in $[-1, 1]$ and dividing an image into $K$ equally sized squares, each square has a side-length of $2/\sqrt{K}$. Let the mask value decrease to $0.5$ at the intersection of the square and the circular isocontour, i.e., at a distance of $1/\sqrt{K}$ from the centre of the kernel as illustrated in Figure 8. Solving this for each kernel in Equation (3) leads to

$$\psi\left(0, 1/\sqrt{K}\right) = 0.5 \iff \sigma_G^{-1} = K \ln 2, \quad \sigma_L^{-1} = \sqrt{K} \ln 2, \quad \sigma_E^{-1} = K/2. \tag{8}$$

Examples of the initial masks obtained when running the IC-SBP with the proposed initialisations are illustrated in Figure 9.

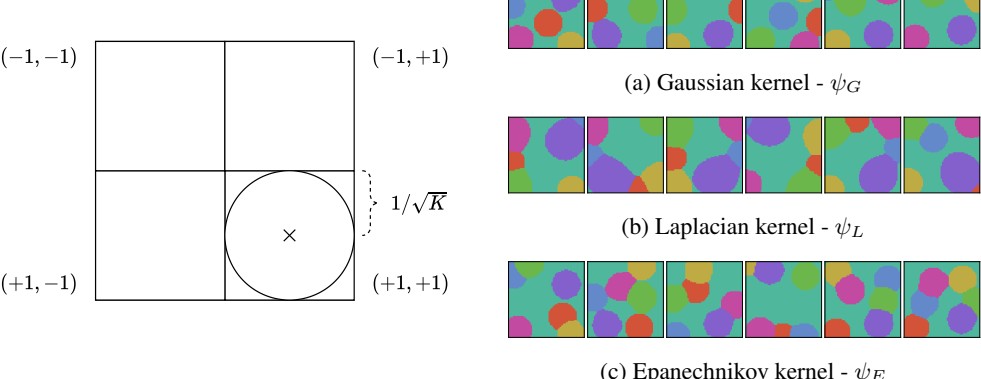

(a) Gaussian kernel - $\psi_G$

(b) Laplacian kernel - $\psi_L$

(c) Epanechnikov kernel - $\psi_E$

Figure 8: Illustration of packing $K = 4$ circular kernels into a square image and linear relative pixel coordinates in $[-1, 1]$, resulting in circular isocontours of radius $1/\sqrt{K}$.

Figure 9: Initial masks obtained when running the IC-SBP with different randomly sampled seed scores, using the initialisations in Equation (8) and $K = 7$.

## D   Datasets

We evaluate GENESIS-V2 on simulated images from ObjectsRoom [44] and ShapeStacks [45] as well as real-world images from Sketchy [46] and APC [47]. ObjectsRoom and ShapeStacks are well established in the context of this work and we follow the same preprocessing procedures as used in Engelcke et al. [17] and Engelcke et al. [18]. As in these works, the default number of object slots is set to $K = 7$ and $K = 9$ for ObjectsRoom and ShapeStacks, respectively, across all models. This work is the first to train and evaluate models that aim to learn object representations without supervision on Sketchy and APC. We therefore developed our own preprocessing and training/validation/test splits, which are described in detail below. The exact splits that were used will be released along with the code for reproducibility.

**Sketchy** The Sketchy dataset [46] is designed for off-policy reinforcement learning (RL), providing episodes showing a robotic arm performing different tasks that involve three differently coloured shapes (blue, red, green) or a cloth. The dataset includes camera images from several viewpoints, depth images, manipulator joint information, rewards, and other meta-data. The dataset is quite considerable in size and takes about 5TB of storage in total. We ease the computational and storage demands by only using a subset of this dataset. Specifically, we use the high-quality demonstrations from the "lift-green" and "stack-green-on-red" tasks corresponding to a total of 395 episodes, 10% of which are set aside as validation and test sets each. Sketchy also contains episodes from a task that involves lifting a cloth and an even larger number of lower-quality demonstrations that offer a wider coverage of the state space. We restrict ourselves to the high-quality episodes that involve the manipulation of solid objects. The number of high-quality episodes alone is already considerable and we want to evaluate whether the models can separate multiple foreground objects. From these episodes, we use the images from the front-left and front-right cameras which show the arm and the foreground objects without obstruction.

The raw images have a resolution of 600-by-960 pixels. To remove uninteresting pixels belonging to the background, 144 pixels on the left and right are cropped away for both camera views, the top 71 and bottom 81 pixels are cropped away for the front-left view, and the top 91 and bottom 61 are cropped away for the front-right view, resulting in a 448-by-672 crop. From this 448-by-672 crop, seven square crops are extracted to obtain a variety of views for the models to learn from. The first crop corresponds to the centre 448-by-448 pixels. For the other six crops, the top and bottom left, centre, and right squares of size 352 are extracted. Finally, we resize these crops to a resolution of 128-by-128 to reduce the computational demands of training the models. This leads to a total of 337,498 training; 41,426 validation; and 41,426 test images. Examples of images obtained with this preprocessing procedure are shown in Figure 10. The default number of object slots is set to $K = 10$ across all models to give them sufficient flexibility to discover different types of solutions.

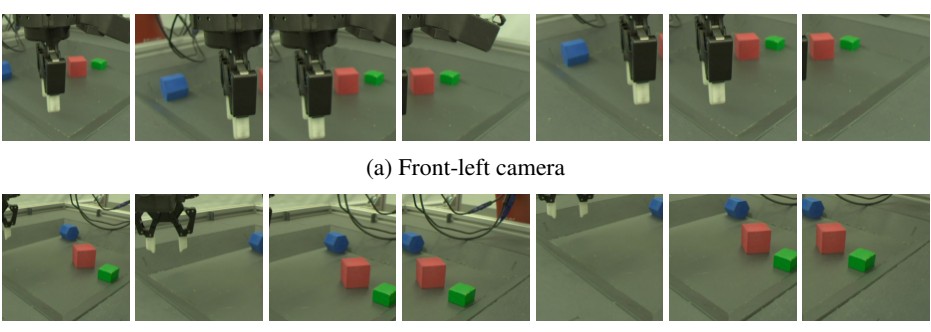

(a) Front-left camera

(b) Front-left camera

Figure 10: 128-by-128 crops as used for training, extracted from the front-left and front-right cameras of a single image from the Sketchy dataset [46]. Showing from left to right: centre, top-left, top-centre, top-right, bottom-left, bottom-centre, and bottom-right crops.

**APC** For their entry to the 2016 Amazon Picking Challenge (APC), the MIT-Princeton team created and released an object segmentation training set, showing a single challenge object either on a shelf or in a tray [47]. The raw images are first resized so that the shorter image side has a length of 128 pixels. The centre 128-by-128 pixels are then extracted to remove uninteresting pixels belonging to the background. Example images after processing are shown in Figure 11. For each object, there exists a set of scenes showing the object in different poses on both the shelf and in the red tray. For each scene, there are images taken from different camera viewpoints. We select 10% of the scenes at random to be set aside for validation and testing each so that scenes between the training, validation, and test sets do not overlap. The resulting training, validation, and test sets consist of 109,281; 13,644; and 13,650 images, respectively. As for Sketchy, the default number of object slots is set to $K = 10$ to provide enough flexibility for models to discover different types of solutions.

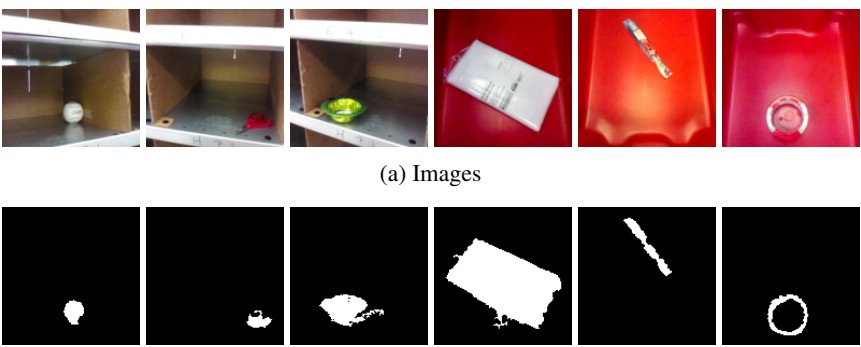

(a) Images

(b) Ground truth segmentation masks

Figure 11: Examples from the APC dataset [47] after cropping and resizing.

# E    Training Details

Models apart from SLOT-ATTENTION are trained with the protocol from Engelcke et al. [17] for comparability, which minimises the GECO objective [56] using the Adam optimiser [68], a learning rate of $10^{-4}$, a batch size of 32, and 500,000 training iterations. The Gaussian standard deviation $\sigma_x$ in Equation (1) is set to $0.7$ and GECO reconstruction goal is set to a negative log-likelihood value per pixel and per channel of $0.5655$ for the simulated datasets and the APC dataset. For Sketchy, a GECO goal of $0.5645$ was found to lead to better segmentations and was used instead. As in Engelcke et al. [17], the GECO hyperparameters are set to $\alpha_g = 0.99$, $\eta = 10^{-5}$ when $C \leq E$ and $\eta = 10^{-4}$ otherwise. $\beta_g$ is initialised to $1.0$ and clamped to a minimum value of $10^{-10}$. For experiments with the auxiliary mask consistency loss in Equation (7), we found that an initial high weighting of the mask loss inhibits the learning of good segmentations, so in these experiments $\beta_g$ is initialised to $10^{-10}$ instead. We refer to MONET trained with GECO as MONET-G to avoid conflating the results with the original settings from Burgess et al. [16]. SLOT-ATTENTION is trained using the official reference implementation with default hyperparameters. Training on 64-by-64 images from ObjectsRoom and ShapeStacks takes around two days with a single NVIDIA Titan RTX GPU. Similarly, training on 128-by-128 images from Sketchy and APC takes around eight days.

# F    Additional results

Table 6 shows a set of ablations for GENESIS-V2 in terms of segmentation performance. A first set of experiments is conducted with an independent prior, the three different distance kernels described in Section 3.2, and semi-convolutional embeddings. The Gaussian kernel appears to perform most robustly and is therefore selected for all other experiments. A second set of experiments is conducted in which models are trained with an auto-regressive prior and either with a semi-convolutional or a standard convolutional output layer for obtaining pixel embeddings. Both the auto-regressive prior and the semi-convolutional operation improve segmentation performance.

Table 6: GENESIS-V2 ablations showing means and standard deviations from three seeds. Highlighting follows an analogous scheme as in Table 1.

| Auto-reg. prior | Kernel | Semi-conv. | ObjectsRoom | | ShapeStacks | |
|---|---|---|---|---|---|---|
| | | | ARI-FG | MSC-FG | ARI-FG | MSC-FG |
| No | $\psi_G$ | Yes | **0.79±0.01** | 0.47±0.17 | **0.79±0.01** | **0.67±0.00** |
| No | $\psi_L$ | Yes | 0.74±0.08 | **0.48±0.20** | **0.79±0.01** | **0.67±0.01** |
| No | $\psi_E$ | Yes | 0.78±0.01 | 0.34±0.08 | 0.78±0.01 | 0.66±0.01 |
| Yes | $\psi_G$ | Yes | **0.84±0.01** | 0.58±0.03 | **0.81±0.00** | **0.68±0.01** |
| Yes | $\psi_G$ | No | 0.79±0.05 | **0.59±0.02** | 0.60±0.38 | 0.56±0.21 |

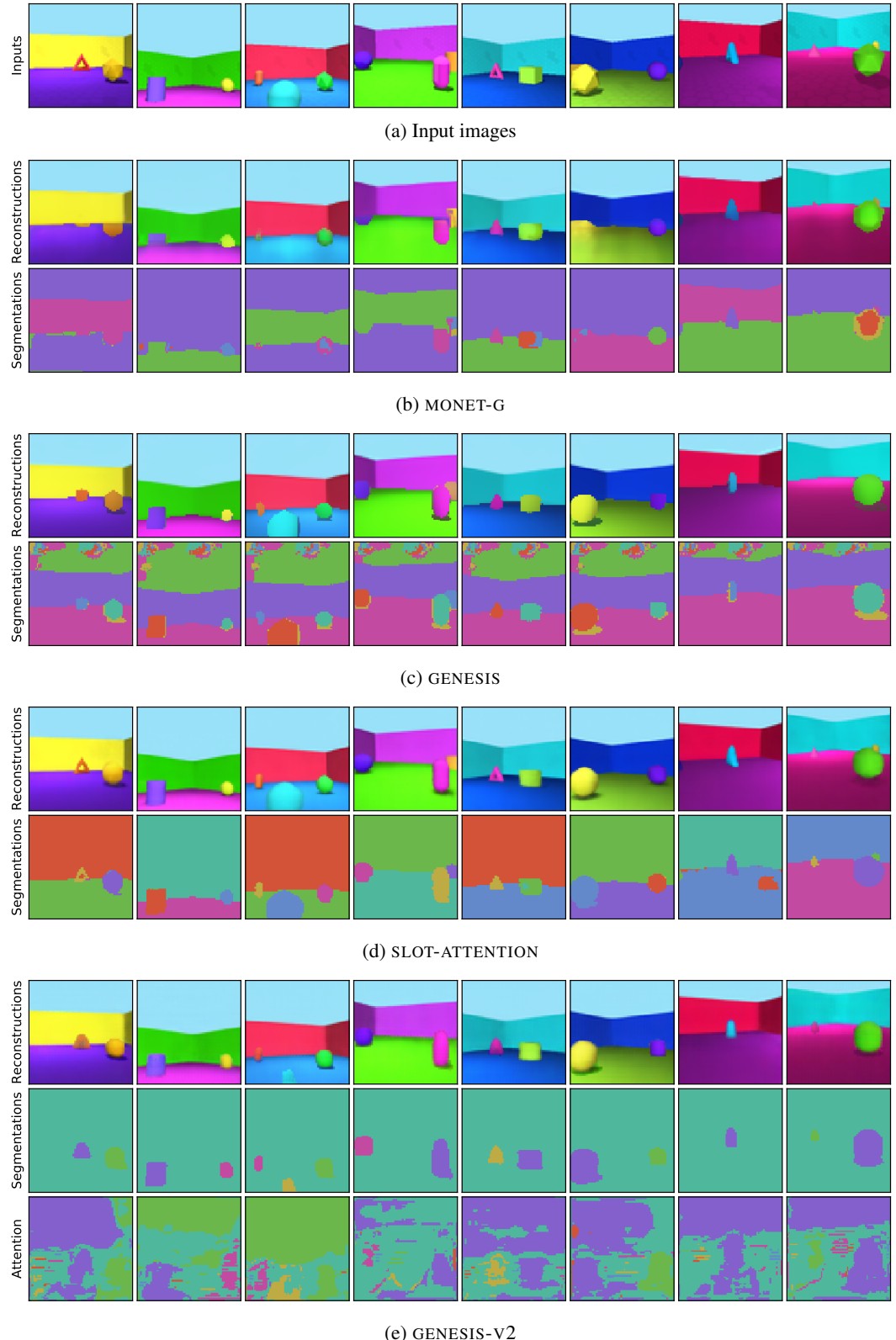

Figure 12: ObjectsRoom reconstructions and segmentations.

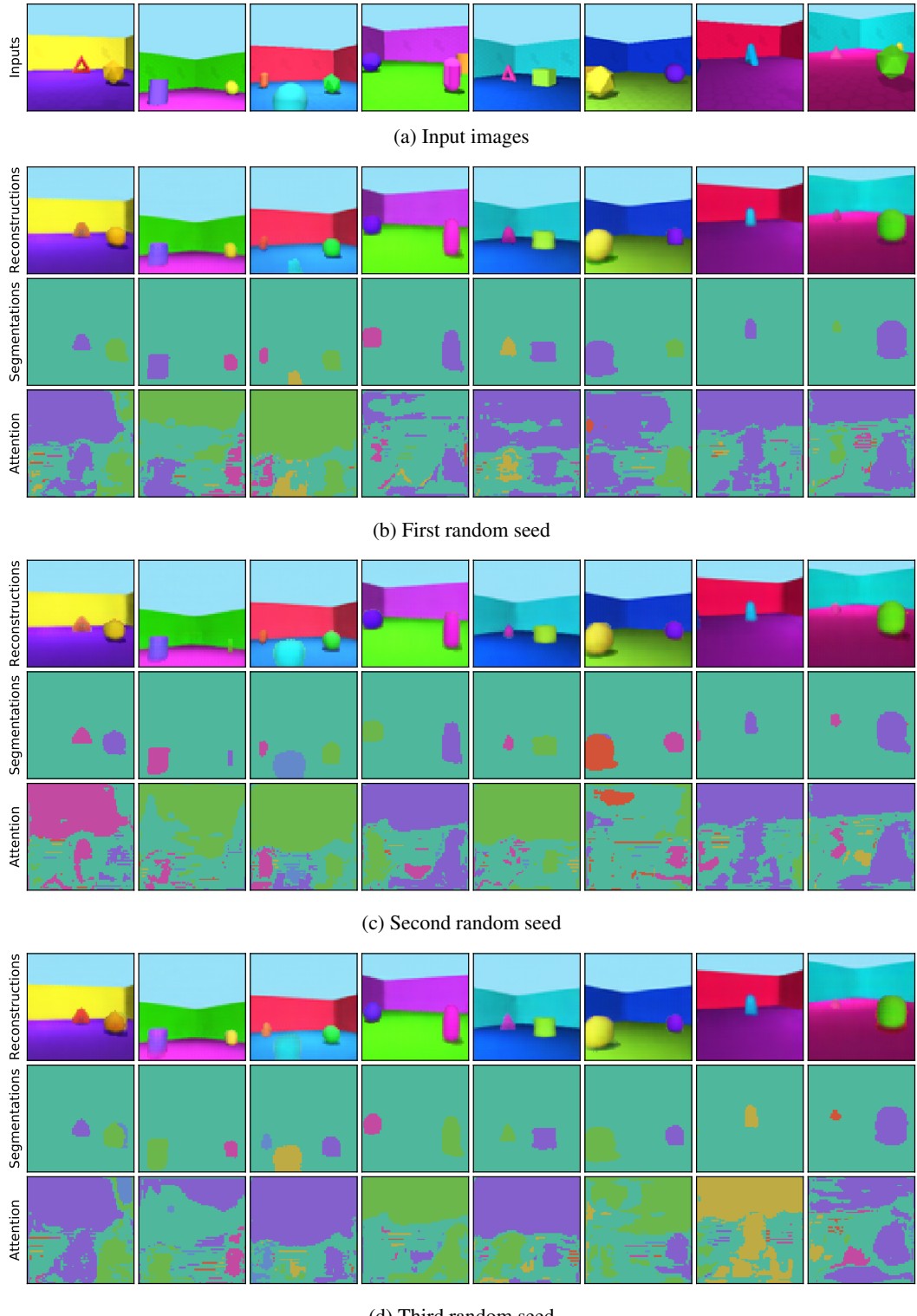

Figure 13: Applying GENESIS-V2 several times to the same images from the ObjectsRoom dataset with three different random seeds shows that the model produces similar reconstructions and segmentations for each seed, but foreground objects are allocated to different slots as indicated by the segmentation colours.

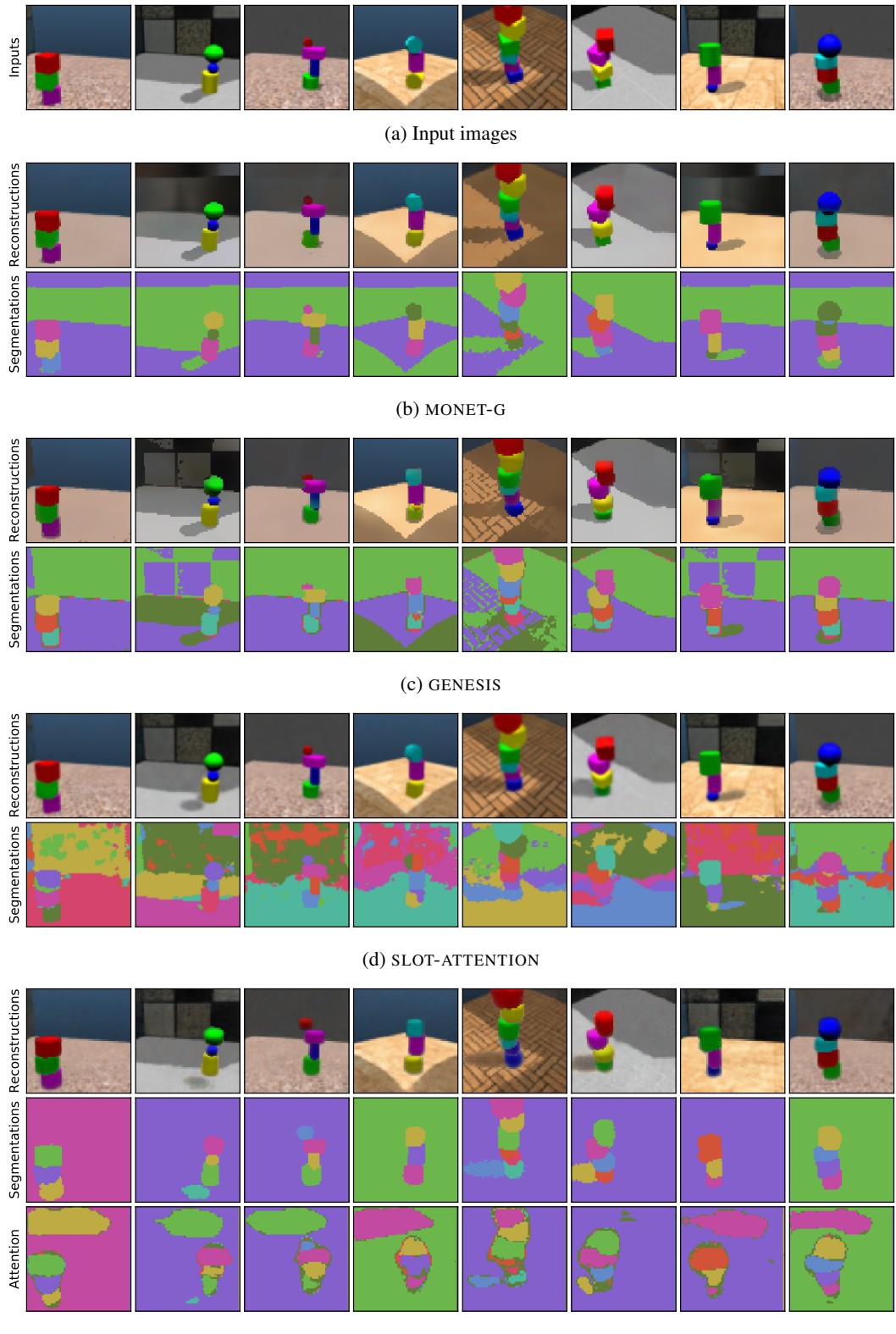

Figure 14: ShapeStacks reconstructions and segmentations.

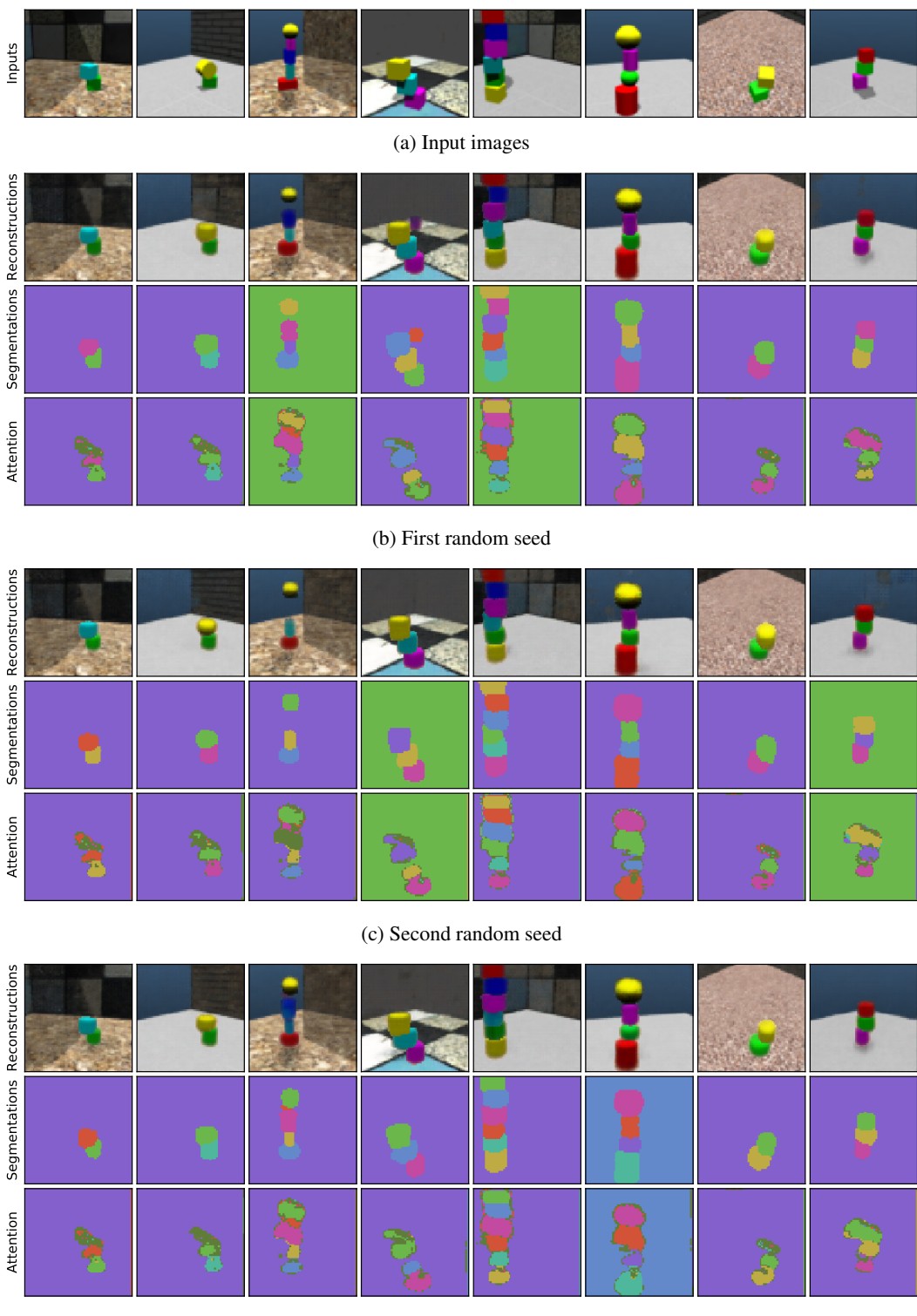

(a) Input images

(b) First random seed

(c) Second random seed

(d) Third random seed

Figure 15: Applying GENESIS-V2 several times to the same images from the ShapeStacks dataset with three different random seeds shows that the model produces similar reconstructions and segmentations for each seed, but components are allocated to different slots as indicated by the segmentation colours.

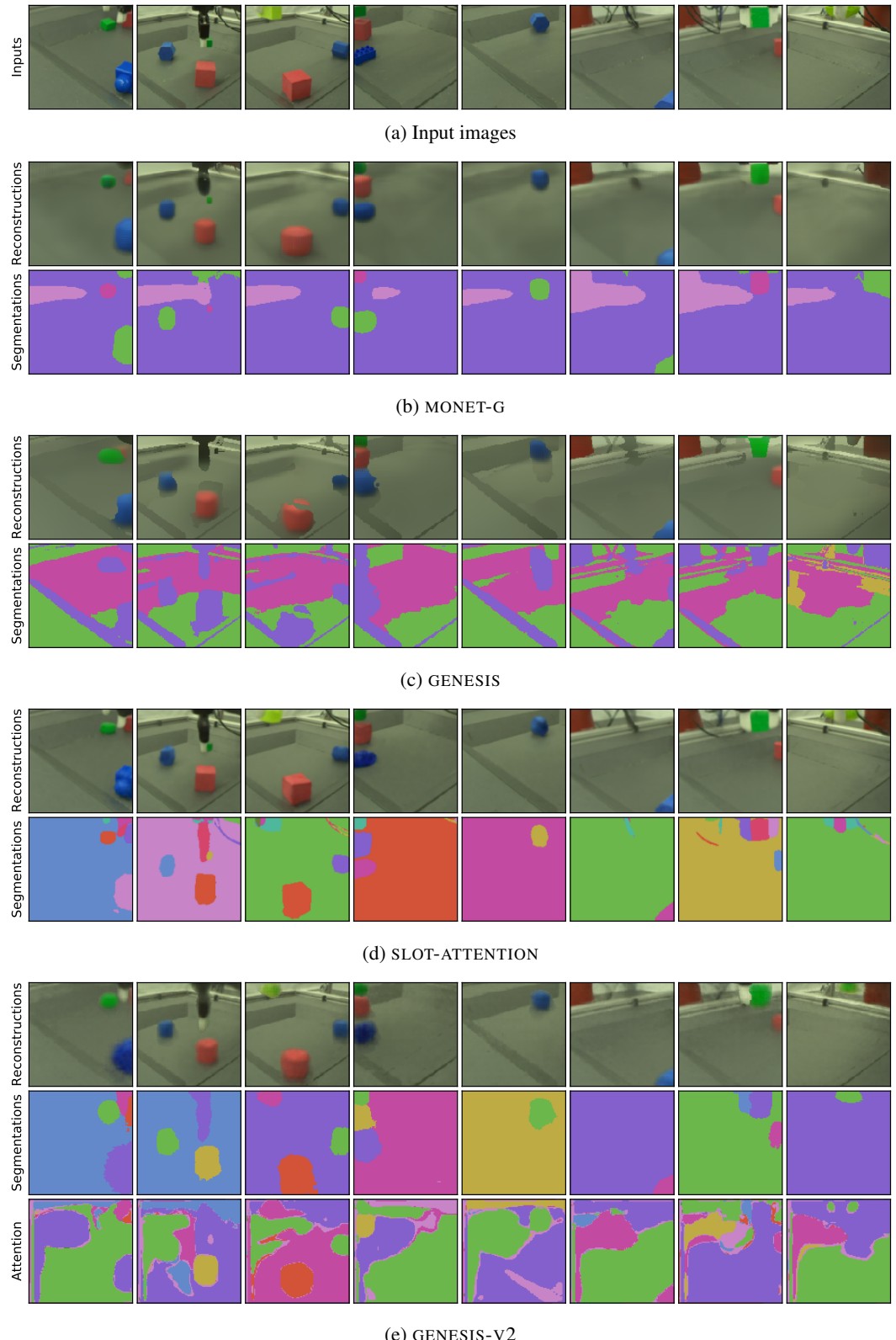

Figure 16: Sketchy reconstructions and segmentations.

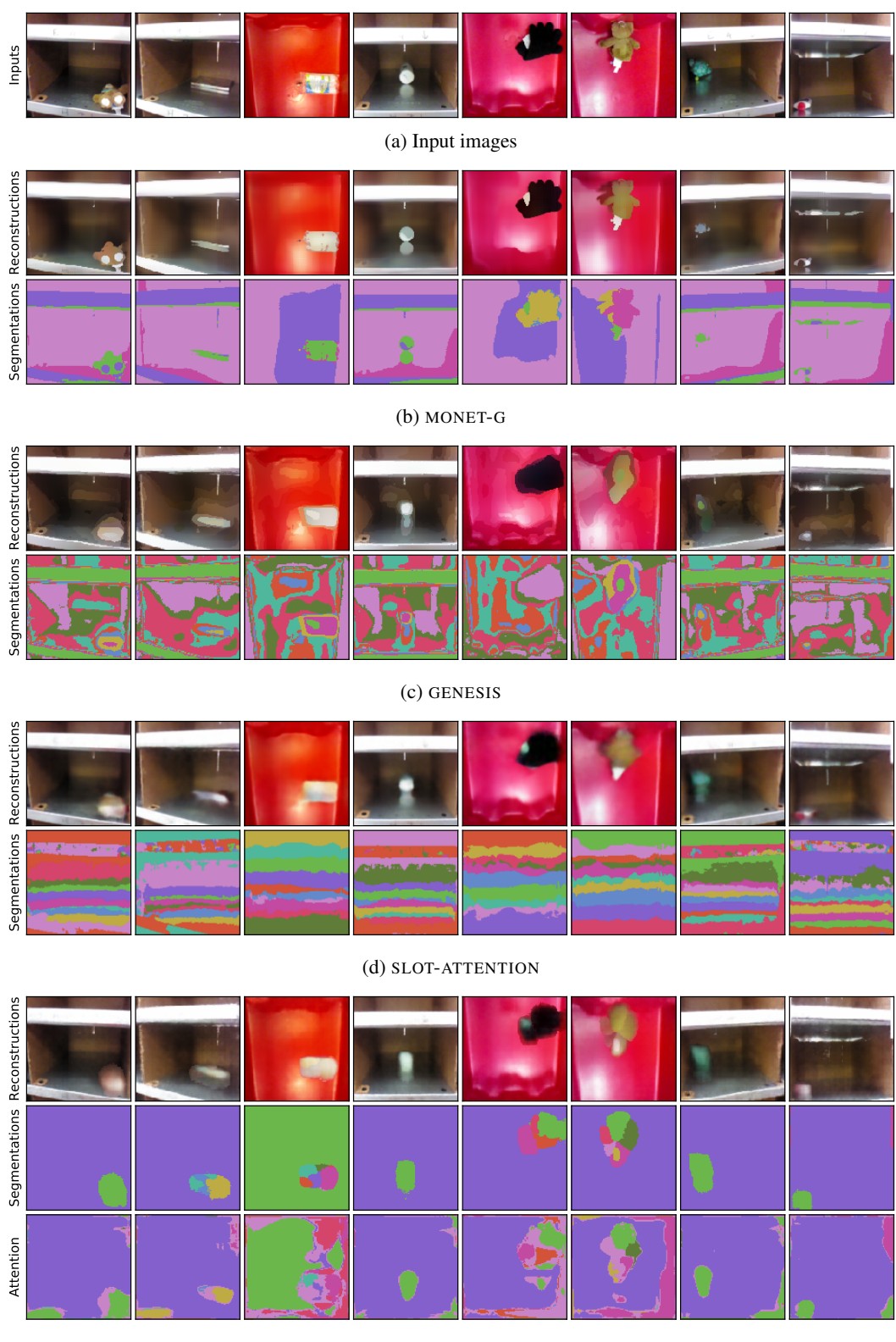

Figure 17: APC reconstructions and segmentations.

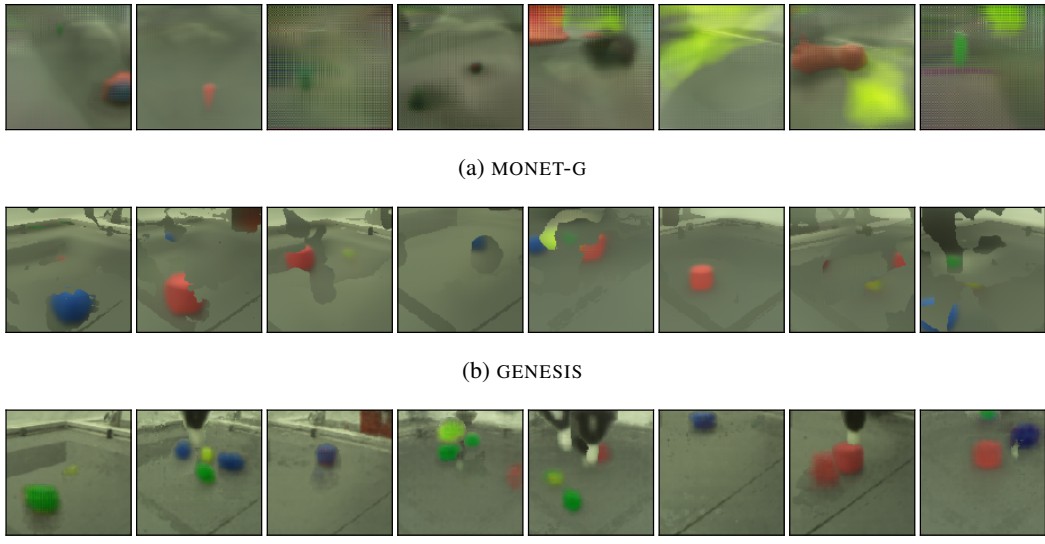

(a) MONET-G

(b) GENESIS

(c) GENESIS-V2

Figure 18: Sketchy samples.

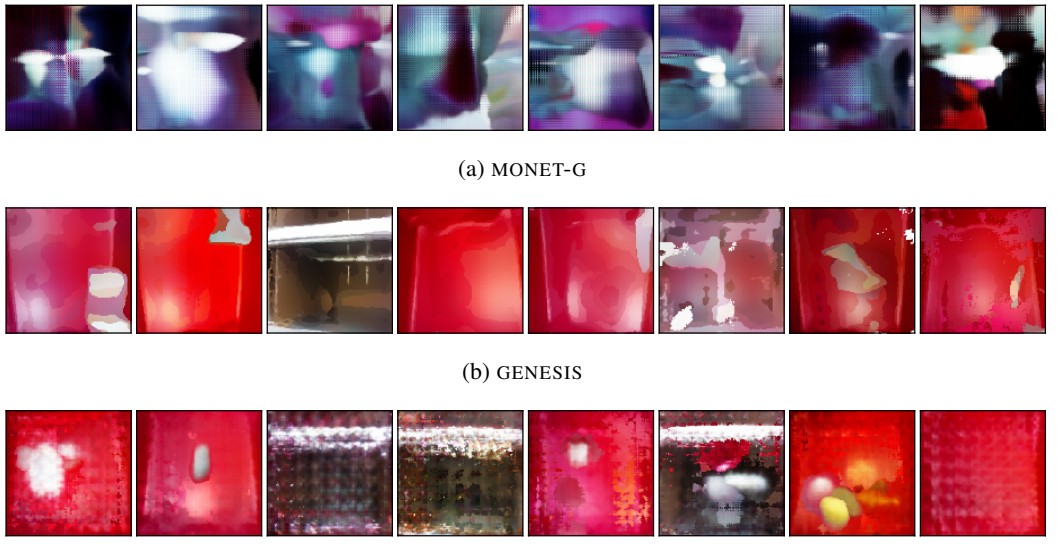

(a) MONET-G

(b) GENESIS

(c) GENESIS-V2

Figure 19: APC samples.

## G    Potential Negative Societal Impacts

GENESIS-V2 is a generative model. Generative models can potentially be used spread disinformation by generating synthetic images for manipulative purposes. At this point in time, however, GENESIS-V2 is only able to generate plausible images when training on simulated images with limited visual complexity. A direct application of this method for malicious purposes is therefore unlikely.

## H    Third-Party Assets

GENESIS-V2 is implemented using PyTorch [69]. In addition to various Python packages, we make use of several third-party assets:

- Kabra et al. [44] (Apache-2.0 License): ObjectsRoom dataset,
- Groth et al. [45] (GPL-3.0 License): ShapeStacks dataset,
- Cabi et al. [46] (Apache-2.0 License): Sketchy dataset,
- Zeng et al. [47] (BSD-2-Clause License): APC dataset,
- Engelcke et al. [17, 18] (GPL-3.0 License): Implementation of GENESIS and MONET-G,
- Locatello et al. [24] (Apache-2.0 License): Implementation of SLOT-ATTENTION,
- Seitzer [60] (Apache-2.0 License): FID computation in PyTorch.

The datasets are publicly available under open-source licenses and consent was therefore not explicitly requested. To the best of our knowledge, none of the datasets contain personally identifiable information or offensive content.