# OpenReview forum: "GENESIS-V2: Inferring Unordered Object Representations without Iterative Refinement"
_NeurIPS.cc/2021/Conference — NeurIPS 2021 Poster_

### Official Review · Reviewer_BKZJ · 2021-07-14

**Rating:** 4
**Confidence:** 4

**Summary:**

This work approaches object-centric decomposition and generative modeling using a combination of prior works, including Genesis [1], Slot Attention [2], semi-convolutional operators for instance segmentations [3], and MONet [4].

**Limitations And Societal Impact:**

The authors show room for improvement in the generation (sampling) of ShapeStacks and APC scenes. They also discuss the limitations of producing a variable number of slots/object masks. It would be great if they could further describe the failure modes of the algorithm or the implications of initial conditions like the seed scores c.

**Main Review:**

Strengths:
- IC-SBP is an interesting combination of existing approaches. It is described reasonably along with the implications of using binary or non-binary masks and possible extensions.
- Multiple metrics (ARI, MSC, and FID) help paint a detailed picture.
- Ablations cover the main tricks used including the mask loss term, semi-conv operators, and auto-regressive prior.
- Decent dataset coverage (including real world data), but nevertheless simplistic and far from game changing.

Questions:
- To what extent is instance coloring related to mean shift clustering? It would be helpful to explore any parallels with the statistics/stochastic processes literature.
- Could you shed some light on the quality of the representations (latents) achieved by Genesis V2? My concern is the spatial pooling used to produce latent statistics (a la Slot Attention) may not be amenable to interpretable features despite the latent bottleneck.
- What is the qualitative effect of sampling different seed scores c on the segmentation outputs?
- What is the rationale of optimizing the scale parameter \sigma of the distance kernel via gradient descent? To what extent did it vary over the course of training in your experiments? Does the value correlate strongly with the number of non-empty masks?

Weaknesses:
- Not enough novelty/exploration of topics of interest.
- One of the claimed advantages (variable number of objects per scene) is not on solid ground, as it doesn’t seem achievable in ObjectsRoom. It requires a heuristic with an additional hyperparameter (number of pixels last explained) even where it works.
- To compare fairly to the baseline models, it is not sufficient to train them using the same protocol, even if it is GECO using a common target reconstruction likelihood. Two things are missing and seem essential:
   (1) A scatterplot showing reconstruction (log LL) versus compression (KL) tradeoffs across the models.
   (2) Results with different reconstruction error targets. Whether a model can achieve a target log LL depends (among other things) on the capacity of its decoder. A particular concern is that MONet-G looks poorly trained on ObjectsRoom. Compared to its performance in the original paper on the same dataset, the way it is trained here suggests a training deficiency.
- Regarding the use of metrics in the field, there’s a claim in the paper which is important but unjustified: ARI “does not penalize the undersegmentation of objects.” Such claims are important in shaping future work in the field and must be properly justified. The only citation here is to [1], where Figure 13 confusingly asserts that “ARI does not penalise the over-segmentation of the foreground objects”. To my knowledge, ARI is a clustering metric which will penalize both under- and over-segmentation. It’s only ARI-FG (which excludes background pixels from the computation) which can lead to a situation like Figure 13 in [1]. It’s unclear whether that figure is computing the ARI for all pixels or just ARI-FG. In the context of this work, it is unclear where ARI-FG will differ from MSC-FG given they’re both excluding background pixels.

Suggestions:
- Figure 1 could be more helpful. Consider labeling the UNet, Conv-GN8-ReLU blocks, etc.
- Consider comparing to a non-parametric segmentation algorithm like QuickShift++ [5] (with a similar number of clusters).
- Move the distance kernel descriptions to the appendix. They’re not used anywhere in the main text.

References:
[1] Engelcke, Martin, et al. "Genesis: Generative scene inference and sampling with object-centric latent representations." arXiv preprint arXiv:1907.13052 (2019).
[2] Locatello, Francesco, et al. "Object-centric learning with slot attention." arXiv preprint arXiv:2006.15055 (2020).
[3] Novotny, David, et al. "Semi-convolutional operators for instance segmentation." Proceedings of the European Conference on Computer Vision (ECCV). 2018.
[4] Burgess, Christopher P., et al. "Monet: Unsupervised scene decomposition and representation." arXiv preprint arXiv:1901.11390 (2019).
[5] Jiang, Heinrich, Jennifer Jang, and Samory Kpotufe. "Quickshift++: Provably good initializations for sample-based mean shift." International Conference on Machine Learning. PMLR, 2018.

**Time Spent Reviewing:**

9

---

> ### Author Response · Authors · 2021-08-09
> **Response to Reviewer BKZJ**
>
> We thank reviewer BKZJ for their time and their feedback. In the following, we attempt to respond to their main questions/criticisms.
>
> Questions:
> - Mean-shift clustering iteratively updates all points so that points belonging to the same "blob" move closer to each other. The final set of clusters is typically obtained with a (non-differentiable) post-processing operation. The IC-SBP could be used to make the aforementioned post-processing operation differentiable.
> - Updating sigma alongside the other parameters gives the model additional flexibility which is likely preferable compared to keeping sigma fixed.
> - We did not conduct a rigorous investigation into the disentanglement of individual latent dimensions, the qualitative behaviour of sampling different seed scores, and how sigma varies during training.
>
> Weaknesses:
> - Lack of novelty/exploration of topics of interest: It would be appreciated if the reviewer could provide a justification for this statement. We believe the development of the IC-SBP is of significant novelty and impact as it allows ideas from instance colouring methods to be leveraged in the context of unsupervised object representation learning.
> - Flexible number of slots: For any method or model, using a flexible number of slots/steps always requires a discrete decision for which a threshold needs to be chosen based on some heuristic. While it is without doubt a weakness that this did not work on the ObjectsRoom dataset, it is clearly demonstrated that Genesis-v2 can extract a variable number of object representations after training on the ShapeStacks dataset with a minimal impact on segmentation performance. Getting object-centric models to behave consistently across datasets of varying appearance without dataset-specific fine-tuning is a general challenge. We believe it is valuable to apply new models to increasingly more difficult datasets and to openly report their weaknesses/limitations/possible areas of improvement.
> - Model comparison: The models trained with GECO all have approximately the same reconstruction error and we believe that quantifying the quality of generated images via FID scores is more insightful than comparing KL divergences. Comparing results with different reconstruction error targets would require a considerable amount of computational resources and it is unclear to us whether significant insights could be gained from this.
> - Segmentation metrics: Thank you for spotting the ambiguity. We meant to write that *ARI-FG* does not penalise the undersegmentation of foreground objects and we will update the paper accordingly. We believe there is a typo in the Genesis paper and that the authors also meant to refer to *under*segmentation. The authors of both IODINE and Genesis refer to ARI-FG simply as "ARI". We intend to avoid this confusion by specifically referring to "ARI-FG" and "ARI". As described in the Genesis paper, there are conceptual differences between ARI-FG and MSC-FG. The latter is asymmetric and does not discard pixels in the predicted segmentation masks that belong to the background according to the ground truth masks.

---

> > ### Comment · Reviewer_BKZJ · 2021-09-01
> > **Sticking with my original rating**
> >
> > Apologies if my comment about lack of novelty came across too harsh. My point is this method could benefit substantially from more rigorous assessment. The justification for using semi-convolutional ops for instance segmentation is well presented in Novotny et al (2018). IC-SBP is an interesting implementation of that idea, but not thoroughly motivated/assessed (e.g. in terms of representation quality, algorithmic stochasticity, etc).
> >
> > My remaining concerns have not been addressed either. Leaving $\sigma$ free to vary, giving the model "additional flexibility," is not how VAEs are typically trained. The effect of sampling different seed scores `c` on the output segmentation needs to be demonstrated. As Reviewer KxNW also pointed out, dynamic stopping (to infer the number of components) needs to be demonstrated more carefully.
> >
> > I hope to see these details in a revised version of the paper.

---

> > > ### Author Response · Authors · 2021-09-05
> > > **Response 2 to Reviewer BKZJ**
> > >
> > > We thank reviewer BKZJ for their response.
> > >
> > > There appear to be two misunderstandings that we would like to clarify:
> > > - The IC-SBP is not an "implementation" of semi-convolutional operators for instance segmentation as developed by Novotny et al.. Instead, it is a differentiable clustering algorithm which is used to make *instance colouring* methods amenable for end-to-end learning without explicit mask supervision. Semi-convolutional operators are a technique for improving the segmentation performance of instance colouring methods. We adopt semi-convolutional operators and derive a principled kernel initialisation to improve the empirical performance of GENESIS-v2. The IC-SBP, however, does not strictly require the use of semi-convolutional operators and quantitative results of training GENESIS-v2 without semi-convolutional operators are included in Table 6.
> > > - Treating the scale parameter \sigma_{G/L/E} in the distance kernel as a learnable parameter is adopted from Novotny et al.. The Gaussian standard deviation \sigma_x in the reconstruction loss term is separate from the scale parameter in the distance kernel and set to a fixed value as commonly done in the literature (see, e.g., MONet, IODINE, and GENESIS).
> > >
> > > Regarding the motivation and assessment of the IC-SBP:
> > > - The use of instance colouring methods in the context of object-centric representation learning and therefore the need for the IC-SBP is motivated by the ability to infer unordered object representations without having to instantiate a predetermined number of slots as required by models that utilise iterative refinement.
> > > - While showing, e.g., the effect of different random seeds on the IC-SBP output might be a nice supplementary illustration, the strong empirical performance across four datasets and the associated qualitative results already demonstrate that the sampling procedure used by the IC-SBP does not introduce prohibitive noise artefacts and that slot allocation does not follow a fixed sequential strategy as produced by models that use RNNs for inference. This confirms that the IC-SBP is indeed suitable for bridging instance colouring methods and unsupervised object-centric models.
> > > - Similarly, while latent traversal illustrations, e.g., might also be a nice supplement, it can already be seen that GENESIS-v2 learns informative and well-compressed representations as demonstrated by the quality of the images generated by sampling from the prior and the quantitative performance in terms of FID scores.

---

### Official Review · Reviewer_KxNW · 2021-07-15

**Rating:** 5
**Confidence:** 4

**Summary:**

This paper proposes a new object-centric generative model called GENESIS-v2. Unlike prior RNN-based approaches using sequential attention, it yields an unordered collection of object representations; and unlike "clustering approaches" based on iterative refinement it does in principle not require fixing the number of slots a priori. This is achieved by extending GENESIS (Engelcke et al., 2020) with a differentiable clustering method based on an Instance Colouring Stick-Breaking Process (IC-SBP).

Here, coordinate-aware (i.e. semi-convolutional) pixel-level embeddings are clustered iteratively by first randomly selecting a cluster seed and then comparing its associated pixel-embedding to all other pixel-embeddings using a learnable distance kernel to extract a mask. When the masks are binary these masks are uniquely defined up to their ordering. A heuristic can be used to decide when to terminate the clustering process and assign the remaining image content to a "background" mask. Once attention masks are obtained, they can be used to derive the associated feature vectors to obtain slot representations, which can be decoded separately as in prior work (eg. IODINE/Slot-Attention). GENESIS-v2 is evaluated on a number of synthetic and more real-world datasets, where it is shown to outperform existing approaches in terms of clustering and generative capabilities.

**Limitations And Societal Impact:**

There is some discussion about the limitations of this model throughout the text, which is sufficient in my view. The societal impact statement is rather brief and generic, but which is also somewhat understandable for this line of work.

**Main Review:**

This paper presents an interesting alternative to the two common paradigms explored in object-centric models for representation learning. A U-net style auto-encoder computes pixel-level embeddings, which are made coordinate-aware by adding their relative pixel coordinates to two dimensions of the embedding as proposed in Novotny et al. (2018). A sequential Instance Colouring Stick-Breaking Process (IC-SBP) then uses these embeddings to derive attention masks from which object representations can be computed similar to in prior work. In particular, the attention-masks are combined with the pixel-embeddings and pooled (and transformed) to obtain parameters of a factored posterior distribution from which object representations can be obtained. Decoding is as in Slot-Attention (Locatello et al., 2020) using separate decoders for each so, which output RGB and a segmentation logit, which can be normalized to obtain a masking. Training proceeds via the Generalized ELBO with Constrained Optimization (GECO) algorithm from Rezende et al. (2018). Additionally an "auxiliary mask loss" can be added to force the attention masks and the segmentation masks to be similar.

The most interesting and novel aspect of GENESIS-v2 is therefore the IC-SBP for clustering.
The IC-SBP makes use of a kernel distance function for comparing pixel embeddings, and it is shown how a meaningful initialisation of their parameters (which are learned) can be derived to yield an initial clustering that incorporates a spatial inductive bias (Figure 9 in Appendix C). Further, although the ordering of the cluster seeds in the IC-SBP is stochastic, it is argued that when the masks are binary, the masks are indeed uniquely defined. This is important, since otherwise the masks may depend on the order in which the cluster seeds were selected, which is undesirable as in the case of RNN-based approaches. However, actually using such discrete mask values is not explored and all experiments are conducted using continuous mask values as far as I can tell. Another potential advantage of using IC-SBP is that one can in principle dynamically decide when to terminate the clustering process using a heuristic (and thus how many slots are obtained) as opposed to committing to a fixed upper bound beforehand. However, here again the paper falls short, as no attempt is made at deriving a general/useful heuristic. Rather, only a naive heuristic to terminate IC-SBP when the sum of the attention mask values < 20 is explored, but which is (understandably) observed to perform poorly on other data sets. Finally, this heuristic is only explored during inference, while training always proceeds with a fixed K as in prior methods.

The experiments are extensive and GENESIS-v2 is compared against GENESIS, MONet, and Slot-Attention on both synthetic and more real-world data. Although the former two baselines are trained using the GECO objective and benefit from the hyper-parameter tuning done for GENESIS-v2, it appears that no further tuning was done for Slot-Attention, which seems somewhat unfair for the more real-world datasets. Regardless, on the synthetic datasets it is convincingly shown how GENESIS-V2 is often the best performing model, when using a fixed K, both in terms of generation (FID) and clustering (ARI & MSC). A metric to measure reconstruction quality such as MSE or nll is strangely absent, which would allow for a better comparison in terms of how well GENESIS-v2 is able to model the data. Currently it seems that the reconstructions by GENESIS-v2 are quite poor compared to say Slot-Attention. For example, in Figure 12 on ObjectsRoom it can be seen how it misses entire objects (4th panel from left) and are quite blurry such as in Figure 5. I agree with the authors that the generative results show room for improvement, since although GENESIS-v2 improves in terms of FID, the absolute FID is still quite high and the image quality is generally quite poor.

On the more real world datasets similar observations hold true and GENESIS-v2 is generally the best performing model. However, here I was quite surprised that K=10 is used for Sketchy even though the images only contain 1 gripper + 3 objects. Similarly, for APC the images contain only a single object, yet K=10 is used. This is significantly different for the hyper parameter choices on the synthetic datasets, where far more objects can be observed and yet K < 10. The reason that I am bringing this up is because baseline methods may be disadvantaged by this choice. For example, Slot-Attention requires all slots to participate in the iterative grouping process, and thus a large K will necessarily yield in more competition among the components. Clearly, it can be argued that this is a disadvantage of such methods and an advantage of GENESIS-v2 that it does not suffer from this. However, in that case, I do expect a set of experiments with a much more reasonable K to verify that baseline methods suffer indeed only because of the misspecification (and perform well if K is set more accurately). I would recommend K = 5-7 for Sketchy and K = 2-4 for APC.

In summary, although I find that the contribution is interesting and a promising direction for future work on object-centric generative models, I can not yet recommend it for acceptance. Indeed, the novel clustering approach proposed here is mainly motivated as (1) leading to randomly ordered object representations without dependencies between them, and (2) not needing to set a fixed number of clusters a priori. However, (1) requires evaluating IC-SBP with discrete masks, which is not done, and (2) is insufficiently explored: no attempt is made at developing a general heuristic, no experiments are provided when training using the current heuristic, etc. I should note that I agree that it has been shown that GENESIS-v2 behaves well when K is misspecified, but this is a lot less significant and not how the current approach is motivated. Further, the experimental evaluation needs to be improved: a measure of reconstruction quality should be provided, equal tuning needs to be applied to all baselines, and on the more real-world datasets experiments need to be included with a more reasonable choice of K.

**Minor Comments**

* It would be good to clarify the "relative" in "relative pixel coordinates". From Figure 8 in the appendix I assumed that these coordinates are relative to the image center?

* It took me a while to figure out how the MSC metric is computed and a ref to the paper where this is described (I believe it is derived from Arbelaez et al. 2010?)

* I think the results in Figure 9 are quite nice and would suggest moving that to the main text.

* Similarly, I would suggest moving some of the technical detail regarding decoding, etc. to the main text and integrate it into the discussion. Currently, the description in the main text is quite high-level for those not familiar with prior work in this area.

* I was a little confused at the naming initially since the decoder is more like Slot-attention / IODINE rather than GENESIS. Perhaps this model deserves its own name?

* Please take a moment to go through the references to correctly cite prior work that has been published (as opposed to citing the arxiv version)

**Post-rebuttal Update**

I think that this paper has a lot of potential, but is currently borderline mainly because the authors fail to provide sufficient empirical support regarding dynamically inferring the correct number of components. In particular:

* No more general heuristic is developed as a stopping condition, which leaves it unclear what stopping condition should be to obtain good performance w.r.t. dynamic component selection
* No experiments are provided where training proceeds using a dynamic number of objects, which leaves it unclear how this affects performance.

There are also other issues regarding empirical evaluation, such as how K is chosen for the baselines and the lack of reporting reconstruction error (or nll). I have corresponded with the authors regarding these issues, and there is a clear disagreement about the importance of these. One concern I initially had was unfounded due to a misunderstanding on my part and was clearly addressed. Unfortunately, there appears to be no interest in addressing the other issues, even though some appear easy to fix. Because of this, I still remain slightly in favor of a borderline reject (5).



**Time Spent Reviewing:**

7

---

> ### Author Response · Authors · 2021-08-09
> **Response to Reviewer KxNW**
>
> We thank reviewer KxNW for their time and the detailed review.
>
> In the following, we attempt to address the concerns that are raised:
> - Discrete vs. continuous attention masks: As correctly summarised, the attention masks returned by the IC-SBP are only "unique up to permutation" if and only if the attention masks are discrete. Otherwise, they are affected by the order in which they are sampled. While inferring representations that are "unique up to permutation" is conceptually interesting and potentially fruitful in other applications, it is not a necessary requirement in this context. The aim of the work and the main motivation of the IC-SBP is to infer object representations that are *unordered*, i.e. without any deterministic ordering. This is satisfied regardless of whether the representations are "unique up to permutation" or not - each object can be allocated to any slot depending on the random seed and it is verified empirically that this works well in practice. Whether or not representations that are "unique up to permutation" work better in this context or otherwise is a potentially interesting avenue for building on the contributions of this paper in future work.
> - Stopping condition: The heuristic for stopping the IC-SBP when inferring a flexible number of slots is a hyper-parameter of the algorithm and it is unclear whether it is possible to derive a generally useful heuristic. Perhaps a heuristic that is slightly more satisfying/intuitive than the one used currently would be to terminate the IC-SBP once 99% of the pixels (or similar) have been allocated to object slots. As mentioned in the paper, training with a flexible number of slots is not practical as it leads to lower GPU utilisation, thus most likely resulting in longer training times.
> - Reconstruction error: We did not include a quantitative analysis of the reconstruction error as (1) it does not directly measure the capabilities that are arguably most interesting (segmentation and generation) and (2) as it is unclear whether much insight can be derived from this. MONet, Genesis, and Genesis-v2 are trained with GECO using the same reconstruction target, so all three models will have approximately the same reconstruction error. SlotAttention is not a VAE and it is trained by minimising the reconstruction error without KL regularisation. It is therefore not surprising that SlotAttention produces visually better reconstructions on most datasets.
> - Number of slots on Sketchy/APC: We intentionally used a larger value of K on Sketchy and APC to be able to observe/explore how different models behave when given a large degree of flexibility. Treating the background as a single entity is not necessarily the "correct" solution as the backgrounds in both datasets could be further partitioned into meaningful sub-components. In all of the considered methods, object slots need to "compete" with each other. It does not seem very likely for the baseline methods to be particularly affected by using “too many” slots during training. Nevertheless, we agree that it would be interesting to see what happens if a smaller number of slots is used during training for these two datasets.
> - Tuning of baselines: While it would be interesting to investigate how far the performance of the baselines can be pushed, this work introduces a novel and conceptually very different mechanism for inferring object representations. Changes in the benchmark results would therefore not significantly affect the core contributions of this work.

---

> > ### Comment · Reviewer_KxNW · 2021-08-18
> > **Clarification requested**
> >
> > Thank you for the reply. Unfortunately, it appears that there is little interest in addressing many of the concerns that I raised and that we disagree on the importance of things like: better-tuned baselines, the importance of lowering K on new datasets, developing a more generally applicable heuristic for inferring the number of objects, analyzing reconstruction error, and training using a variable number of objects.
> >
> > That said, from your reply it seems that my concern about not evaluating the method using binary masks is unfounded and so I would like to ask for clarification. The reason why I considered evaluating binary masks important is because they only allow for a unique ordering, which I believed to be responsible for alleviating the problem of creating spurious dependencies between the object representations (as you argue could be the case for sequential RNN-based approaches). Is not true that in the continuous case, the network could learn to exploit the flexibility of having a stochastic ordering during the IC-SBP to create dependencies between the objects? That is, more so than in the deterministic case where the ordering is pre-determined? If this is indeed a misunderstanding on my part then I agree that this particular issue is no longer problematic.
> >
> > I would also like to comment on the part about reconstruction error in your reply. It is my understanding that the goal of this work is also to learn object _representations_, beyond just focusing on generation and segmentation. From that perspective, it is natural to consider reconstruction error as an additional metric, since it approximately conveys whether information about the segmented objects are correctly captured by the representation (as opposed to only encoding location/shape as is needed for segmentation). If only generation and segmentation are of interest then a comparison to different baselines such as [Copy-Pasting GAN](https://arxiv.org/pdf/1905.11369.pdf), [ReDO](https://arxiv.org/pdf/1905.13539.pdf), etc. becomes more relevant, especially on APC where in essence only foreground-background segmentation is required.

---

> > > ### Author Response · Authors · 2021-08-20
> > > **Response 2 to Reviewer KxNW**
> > >
> > > We thank reviewer KxNW for their reply and for engaging in this discussion.
> > >
> > > Binary vs. continuous masks:\
> > > RNN-based models need to decide on a “strategy” for sequentially segmenting images, e.g., left-to-right, front-to-back, etc. The discovery of such a strategy poses an additional optimisation challenge as it results in a “symmetry breaking problem” while also being susceptible to the typical vanishing gradient problems in RNNs, especially when sequences are very long. The IC-SBP does not need to perform this kind of symmetry breaking as a certain order is randomly instantiated via stochastic sampling. Importantly, any other sampled order is equally valid. Moreover, this “stochastic ordering” also makes it disadvantageous for the network to create and rely on dependencies between iteration steps as it cannot rely on specific scene components having been explained beforehand, e.g., there can never be a fixed left-to-right strategy or similar. We recognise that this might currently not be communicated in sufficient clarity and we will revise the manuscript accordingly.
> > >
> > > Reconstruction error:\
> > > Learning useful object representations is indeed an objective of this work and we argue that this can be well quantified by image segmentation and generation metrics. While the former measures whether objects are being separated, the latter requires both (a) good reconstruction and (b) good compression. In a VAE, the quality of images generated by sampling from the prior is bounded by the quality of the training reconstructions. An increase in the quality of generated images is directly correlated with how informative and compressed the representations are. While reconstruction error can provide an indication of representation quality, it can also be misleading; for example, the identity mapping has a reconstruction error of zero without extracting useful representations. We have therefore been hesitant to report reconstruction errors, as a better reconstruction error does not necessarily equate to better object representations.
> > >
> > > Regarding the reviewer’s “minor comments” in the original review:
> > > - Yes, pixel coordinates are relative to the image centre in our implementation.
> > > - Yes, the MSC is derived from Arbelaez et al. 2010. We will add a reference to this paper.
> > > - Thank you for the suggestion of moving Figure 9 and some of the technical details to the main text. We will try to do so if space is available.
> > > - We decided to name the model GENESIS-v2 as it is most similar in its capabilities to GENESIS. Both models can perform both segmentation and object-by-object image generation, with the latter being facilitated by an autoregressive prior. Neither IODINE nor SlotAttention can perform the latter.
> > > - Thank you for spotting that some references still correspond to arXiv versions. We will update the manuscript accordingly.

---

### Official Review · Reviewer_HXzc · 2021-07-15

**Rating:** 6
**Confidence:** 4

**Summary:**

Authors propose an object-centric generative model of images. This is derived from GENESIS, but replaces iterative inference with a clustering approach inspired by recent instance segmentation techniques. Results on two synthetic and two real datasets show significantly better performance than recent methods GENESIS and Slot Attention.

**Limitations And Societal Impact:**

Yes, both fine.

**Main Review:**

Strengths:

The paper is generally clear and well-written. Components of the model are motivated reasonably and described comprehensibly.

The model itself is reasonably simple and elegant -- particularly the idea of using an embedding-clustering approach to allow pixels to become grouped in a more 'bottom-up' fashion than existing methods. This approach is novel within the field of structured generative modelling, but inspired by existing techniques in instance segmentation.

The evaluation is reasonably comprehensive, and includes some real-world (albeit rather tightly controlled) data, which is welcome. Results show the method out-performs GENESIS and Slot Attention, by varying degrees, on most datasets, for both generation and segmentation/decomposition tasks.

Concerns / suggestions / omissions:

The technical contribution fairly small. In particular, the model combines ideas from GENESIS, MONet, and clustering-based segmentation, but it doesn't do anything radically new.

155: what is the stopping condition? It seems this could be important, as it affects how likely the model is to over-/under-segment

Fig.2: related to the previous, please discuss what inductive bias of the proposed method means it prefers to treat floor+walls as one object? There is not really anything in the data that means this is the 'correct' segmentation (even though it might look correct to a human)

Tab.1: the bolding here seems inconsistent with the standard deviations -- it looks from the numbers like GENESIS is statistically closer to GENESIS-V2 than the bolding would imply?

130: the described model does not seem to fit the usual structure of a stick-breaking process -- or at least, no more than MONet does --; maybe justify this description, or think of something more appropriate? (this may relate to the previous point; if the stopping condition were that some bernoulli rv is 1, the name would be more apt)

"Generative modeling of infinite occluded objects for compositional scene representation" [Yuan, ICML 19] should be cited.

35: why is GENESIS particularly interesting vs. the many other models cited?

113: formulating an inference model seems to imply use of SGVB, which has not been stated yet

---

## Post-rebuttal

The authors have addressed several of my concerns in their comment, and I remain somewhat in favor of acceptance. The technical contribution (the IC-SBP itself) is relatively small, but still of interest. The issues raised by KxNW and BKZJ are of some concern, but seem to me well-enough addressed by the respective author responses.

**Time Spent Reviewing:**

2.5

---

> ### Author Response · Authors · 2021-08-09
> **Response to Reviewer HXzc**
>
> We thank reviewer HXzc for their time and the thoughtful feedback. Below we respond to the concerns that are raised in the review:
> - We believe the development of the IC-SBP is a significant technical contribution as it allows us to leverage ideas from instance colouring methods in the context of unsupervised object representation learning.
> - l155 stopping condition: A variety of stopping conditions can be used, so we decided to not restrict ourselves to one specific implementation when describing the general method. The stopping condition that we chose to use in some of the experiments is described at l244.
> - Figure 2 (background segmentation): We share the sense that segmenting the entire background as a single component is not always the "correct" segmentation and that there are situations where it would be desirable to segment the floor, walls, sky, shelves, etc. separately. We conjecture that the behaviour of Genesis-v2 arises from the fact that the background structures are of finite variety, i.e., even when viewing only a fraction of the background, it is possible to largely predict the rest of the background. In comparison to the baseline methods, we argue that the observed behaviour is the result of better conceptual compression in terms of modelling the scenes as a whole rather than segmenting the background by pixel colour. This is induced by the KL regularisation during training which encourages efficient compression and penalises redundant information between slots. We will add this discussion to the paper.
> - Table 1: Thanks for spotting; will update.
> - Stick-breaking process: Thank you for the comment; will add an explanation.
> - [Yuan, ICML 19]: Thank you for pointing us to this work; will add to the literature review.
> - "why is GENESIS...": This work takes significant inspiration from Genesis by developing an improved model that can perform both unsupervised object segmentation and object-centric scene generation, with the latter being facilitated by an auto-regressive prior --- hence the name of the model proposed in this work.
> - SGVB: It is described that the model is trained as a VAE which directly implies the use of SGVB.

---

### Official Review · Reviewer_dzPr · 2021-07-17

**Rating:** 7
**Confidence:** 4

**Summary:**

This work proposes a new object centric generative model called genesis-v2. The main contribution is using an Instance Colouring Stick-Breaking Process for assigning pixels to slots. The idea is to randomly select a pixel as seed, assign close pixels based on a kernel, select another random pixel from the remaining unassigned pixels for K steps. Authors claim two advantages, 1: no ordering can be imposed, 2: number of clusters don't need to be set initially. This work shows results on two synthetic datasets and two real world ones and compared to their baseline and slot attention they achieve impressively better results both qualitatively and quantitatively.

**Limitations And Societal Impact:**

Authors indicate some of the limitations in the manuscripts with future work to address them. One of the main limitations is the training speed which makes it infeasible to try on a real segmentation dataset like Pascal os MS CoCo. On the fairly simple real world APC dataset, it takes about 8 days to train.

Another limitation is probably the number of objects it can handle. I would be quite surprised if it can be scaled to handle ~ 100 objects in one scene, like images with pedestrians in busy street scenes. Mainly due to full image pixel level multiplications and attention maps per object, auto regressive prior and the for loop of the colour clustering.

**Main Review:**

The idea of using Instance colouring clustering technique with random seed although is closely related to slot attention but has some novel aspects. Since the slots in slot attention are also randomly initialized the effect is similar to the icsbp proposed here (no specific ordering).

Authors claim that slot attention needs a preset number of slots, whereas they don't. This is not completely true. First, for practical reasons of gpu utilization genesis-v2 also uses a fixed number during training. Their ablation in appendix shows their best results is with a fixed number during inference. So their flexibility claim is not beneficial or advantageous at least in this dataset. In the objectRoom the flexible inference is even disadvantageous. As explained in slot attention, they also can adapt to a flexible number of objects at inference. I would argue that slot attention also fixes the number of slots during training for training efficiency and better parallelization. So there is no restriction in that regard.

For the distance kernel, authors add the pixels position to two dimensions of the embedding and then usually use a gaussian distance kernel. Interestingly they derive an initial value for the sigma based on the set K (number of slots). Again it goes against the flexibility claim, assuming a set K. Deriving the initial sigma based on equal initial mass assignment is interesting and useful.

The most impressive aspect was the experimental results. Qualitatively, they have almost perfect background segmentation. Typically, unsupervised segmentation masks tend to over segment background. This aspect of their results is intriguing to me. There is not much discussion on why genesis-v2 is better at recognizing background. An ablation on K during training can potentially provide insight. My question is would it still segment background/foreground with a larger K, such as 32 instead of 9.

----POST Rebuttal
The idea presented here is interesting and experiments show it most probably helps with symmetry breaking and avoiding bad local minimas. Other reviewers concerns like training all with Geco and Monet-g not being trained properly has probably merit but the results in Genesis-v2 are better in my experience. Finetuning the baselines better most probably will decrease the gap but Genesis-V2 has legitimately  good results.
Therefore, this paper is contributing to the community in my opinion which leans me toward acceptance.
It would still be helpful to include results for a larger K such as 32 to showcase the applicability of this model with larger variety of number of objects in each scene.

**Time Spent Reviewing:**

5

---

> ### Author Response · Authors · 2021-08-09
> **Response to Reviewer dzPr**
>
> We would like to thank reviewer dzPr for their time as well as the thoughtful and detailed feedback. Below, we respond to several points raised by the reviewer:
> - Fixed vs. flexible number of slots: In Table 2, we compare Genesis-v2 with a fixed vs. flexible number of slots on the ShapeStacks dataset. While there is a small drop in segmentation performance for the base model, it can be seen that the difference when using the proposed mask consistency loss is marginal, which is achieved with a significantly smaller average number of slots. We advocate for the ability to use a flexible number of slots to improve computational efficiency, not necessarily to further improve segmentation performance. In contrast to Genesis-v2, models that use iterative refinement (SlotAttention/IODINE) strictly require an initial instantiation of a fixed number of slots. While the authors of SlotAttention show that the number of slots can be over-specified so that "empty" slots can be ignored *after clustering*, we argue that this is (1) computationally wasteful and (2) fails for inputs where the number of objects in the input is, perhaps unexpectedly, larger than the number of instantiated slots.
> - Distance kernel scale: We believe it is reasonable to initialise sigma at the beginning of training based on prior knowledge about the dataset or the typical size of objects. This does not affect the ability of Genesis-v2 to use a flexible number of slots after training. Other initialisation schemes that are independent of K (i.e. the number of slots used during training) might prove similarly or possibly even more effective.
> - Background segmentation: We conjecture that the behaviour of Genesis-v2 arises from the fact that the background structures are of finite variety, i.e., even when viewing only a fraction of the background, it is possible to largely predict the rest of the background. Specifically, we argue that the observed behaviour is the result of better conceptual compression in terms of modelling the scenes as a whole rather than segmenting the background by pixel colour. This is induced by the KL regularisation during training which encourages efficient compression and penalises redundant information between slots. We will add this discussion to the paper.

---

### Decision · Program_Chairs · 2021-09-28

**Decision:**

Accept (Poster)

**Comment:**

The paper proposes an approach to object learning that performs well on two synthetic and two real-world benchmarks. Even after the discussion phase, the reviewers could not agree on the evaluation of the paper: two of them are in favor of acceptance, while two lean towards rejection. After reading the reviews, the author responses, the discussion, and the paper itself, below is a summary of some key points.

Pros:
1) Reasonable model, which, interestingly, potentially is less rigid in terms of pre-fixing the number of object slots than previous models
2) Good empirical performance on two real-world and two synthetic datasets. Evaluation on real-world datasets is a step in the right direction.

Cons:
1) There are doubts about the evaluation being fair to the baselines in terms of hyperparameter tuning. In particular the number of slots on the real-world datasets seems potentially mis-specified. (it is great that the proposed model can deal with this, but trying baselines with the appropriate number of slots is still a crucial experiment)
2) The paper combines ideas from several prior lines of work - GENESIS, Slot Attention, Novotny et al., stick-breaking processes. It is unclear which components actually make it work. There are some ablation studies, but they are performed on synthetic datasets and almost none of them show any significant change.  On all datasets except for one the model works very similar to Slot Attention. The only big difference is on the APC dataset, but it is very unclear where it comes from. The KL penalty? The IC-SBP? The misspecified K in slot attention? Some of the other training parameters? It is really unclear.
3) The claims about adaptive number of objects should be clarified - while I believe there is substance to them, it is apparent from the reviewers' comments that the current presentation can be confusing.

To summarize, the paper presents a well-working unsupervised object segmentation system, but there are some concerns about comparison to the baselines and there is lack of analysis of the reasons for the good performance of the model. The latter point in my opinion is crucial, since without that it is impossible for other researchers to make good use of the paper. Papers should be not (just) about showing off SOTA results, but about communicating clear and usable insights. Thus, at this point I recommend rejection.

(The work does look promising and I encourage the authors to take the feedback into account and re-submit to a different venue. The resistance of the authors to the comments/requests from some of the reviewers is a bit surprising and does not serve them well)

**Consistency Experiment:**

NeurIPS has a long history of experimentation. In 2014, NeurIPS ran an experiment in which 10% of submissions were reviewed by two independent committees to quantify the randomness in the review process. This year, we repeated a variant of this experiment to see how the quality of the review process has changed over time.  This paper was part of the experiment and was therefore assigned to two committees (consisting of reviewers, an Area Chair, and a Senior Area Chair) that reached independent decisions.  If both committees made the same recommendation, this recommendation was followed. If a single committee recommended acceptance, the paper was accepted (with the exception of a few cases in which the other committee identified what we considered a fatal flaw, e.g., an error in a key result).

This copy’s committee reached the following decision: **Reject**

The other committee assigned to the paper recommended **Accept (Poster)**.  You can find the other set of reviews, along with any follow up discussion with the authors here:
https://openreview.net/forum?id=ws4BkjI1l-q